# Global Optimality of Local Search
# for Low Rank Matrix Recovery

**Srinadh Bhojanapalli**
srinadh@ttic.edu

**Behnam Neyshabur**
bneyshabur@ttic.edu

**Nathan Srebro**
nati@ttic.edu
Toyota Technological Institute at Chicago

## Abstract

We show that there are no spurious local minima in the non-convex factorized parametrization of low-rank matrix recovery from incoherent linear measurements. With noisy measurements we show all local minima are very close to a global optimum. Together with a curvature bound at saddle points, this yields a polynomial time global convergence guarantee for stochastic gradient descent *from random initialization*.

## 1 Introduction

Low rank matrix recovery problem is heavily studied and has numerous applications in collaborative filtering, quantum state tomography, clustering, community detection, metric learning and multi-task learning [21, 12, 9, 27].

We consider the "matrix sensing" problem of recovering a low-rank (or approximately low rank) p.s.d. matrix[1] $\boldsymbol{X}^* \in \mathbb{R}^{n \times n}$, given a linear measurement operator $\mathcal{A} : \mathbb{R}^{n \times n} \to \mathbb{R}^m$ and noisy measurements $\boldsymbol{y} = \mathcal{A}(\boldsymbol{X}^*) + \boldsymbol{w}$, where $\boldsymbol{w}$ is an i.i.d. noise vector. An estimator for $\boldsymbol{X}^*$ is given by the rank-constrained, non-convex problem

$$\underset{\boldsymbol{X} : \text{rank}(\boldsymbol{X}) \leq r}{\text{minimize}} \quad \|\mathcal{A}(\boldsymbol{X}) - \boldsymbol{y}\|^2. \tag{1}$$

This matrix sensing problem has received considerable attention recently [30, 29, 26]. This and other rank-constrained problems are common in machine learning and related fields, and have been used for applications discussed above. A typical theoretical approach to low-rank problems, including (1) is to relax the low-rank constraint to a convex constraint, such as the trace-norm of $\boldsymbol{X}$. Indeed, for matrix sensing, Recht et al. [20] showed that if the measurements are noiseless and the measurement operator $\mathcal{A}$ satisfies a restricted isometry property, then a low-rank $\boldsymbol{X}^*$ can be recovered as the unique solution to a convex relaxation of (1). Subsequent work established similar guarantees also for the noisy and approximate case [14, 6].

However, convex relaxations to the rank are not the common approach employed in practice. In this and other low-rank problems, the method of choice is typically unconstrained local optimization (via e.g. gradient descent, SGD or alternating minimization) on the factorized parametrization

$$\underset{\boldsymbol{U} \in \mathbb{R}^{n \times r}}{\text{minimize}} \, f(\boldsymbol{U}) = \|\mathcal{A}(\boldsymbol{U}\boldsymbol{U}^\top) - \boldsymbol{y}\|^2, \tag{2}$$

where the rank constraint is enforced by limiting the dimensionality of $\boldsymbol{U}$. Problem (2) is a non-convex optimization problem that could have many bad local minima (as we show in Section 5), as well as saddle points. Nevertheless, local optimization seems to work very well in practice. Working on (2) is much cheaper computationally and allows scaling to large-sized problems—the number of optimization variables is only $O(nr)$ rather than $O(n^2)$, and the updates are usually very cheap, especially compared to typical methods for solving the SDP resulting from the convex relaxation. There is therefore a significant disconnect between the theoretically studied and analyzed methods (based on convex relaxations) and the methods actually used in practice.

Recent attempts at bridging this gap showed that, some form of global "initialization", typically relying on singular value decomposition, yields a solution that is already close enough to $\boldsymbol{X}^*$; that local optimization from this initializer gets to the global optima (or to a good enough solution). Jain et al. [15], Keshavan [17] proved convergence for alternating minimization algorithm provided the starting point is close to the optimum, while Zheng and Lafferty [30], Zhao et al. [29], Tu et al. [26], Chen and Wainwright [8], Bhojanapalli et al. [2] considered gradient descent methods on the factor space and proved local convergence. But all these studies rely on global initialization followed by local convergence, and do not tackle the question of the existence of spurious local minima or deal with optimization starting from random initialization. There is therefore still a disconnect between this theory and the empirical practice of starting from random initialization and relying *only* on the local search to find the global optimum.

In this paper we show that, under a suitable incoherence condition on the measurement operator $\mathcal{A}$ (defined in Section 2), with noiseless measurements and with $\operatorname{rank}(\boldsymbol{X}^*) \leq r$, the problem (2) has no spurious local minima (i.e. all local minima are global and satisfy $\boldsymbol{X}^* = \boldsymbol{U}\boldsymbol{U}^\top$). Furthermore, under the same conditions, all saddle points have a direction with significant negative curvature, and so using a recent result of Ge et al. [10] we can establish that stochastic gradient descent from random initialization converges to $\boldsymbol{X}^*$ in polynomial number of iterations. We extend the results also to the noisy and approximately-low-rank settings, where we can guarantee that every local minima is close to a global minimum. The incoherence condition we require is weaker than conditions used to establish recovery through local search, and so our results also ensures recovery in polynomial time under milder conditions than what was previously known. In particular, with i.i.d. Gaussian measurements, we ensure no spurious local minima and recovery through local search with the optimal number $O(nr)$ of measurements.

**Related Work**  Our work is heavily inspired by Bandeira et al. [1], who recently showed similar behavior for the problem of community detection—this corresponds to a specific rank-1 problem with a linear objective, elliptope constraints and a binary solution. Here we take their ideas, extend them and apply them to matrix sensing with general rank-$r$ matrices. In the past several months, similar type of results were also obtained for other non-convex problems (where the source of non-convexity is *not* a rank constraint), specifically complete dictionary learning [24] and phase recovery [25]. A related recent result of a somewhat different nature pertains to rank unconstrained linear optimization on the elliptope, showing that local minima of the rank-constrained problem approximate well the global optimum of the rank unconstrained convex problem, even though they might *not* be the global minima (in fact, the approximation guarantee for the actual global optimum is better) [18].

Another non-convex low-rank problem long known to not possess spurious local minima is the PCA problem, which can also be phrased as matrix approximation with full observations, namely $\min_{\operatorname{rank}(\boldsymbol{X}) \leq r} \|A - X\|_F$ (e.g. [23]). Indeed, local search methods such as the power-method are routinely used for this problem. Recently local optimization methods for the PCA problem working more directly on the optimized formulation have also been studied, including SGD [22] and Grassmannian optimization [28]. These results are somewhat orthogonal to ours, as they study a setting in which it is well known there are never any spurious local minima, and the challenge is obtaining satisfying convergence rates.

The seminal work of Burer and Monteiro [3] proposed low-rank factorized optimization for SDPs, and showed that for extremely high rank $r > \sqrt{m}$ (number of constraints), an Augmented Lagrangian method converges asymptotically to the optimum. It was also shown that (under mild conditions) any rank deficient local minima is a global minima [4, 16], providing a post-hoc verifiable sufficient condition for global optimality. However, this does not establish any a-priori condition, based on problem structure, implying the lack of spurious local minima.

While preparing this manuscript, we also became aware of parallel work [11] studying the same question for the related but different problem of matrix completion. For this problem they obtain a similar guarantee, though with suboptimal dependence on the incoherence parameters and so suboptimal sample complexity, and requiring adding a specific non-standard regularizer to the objective—this is not needed for our matrix sensing results.

We believe our work, together with the parallel work of [11], are the first to establish the lack of spurious local minima and the global convergence of local search from random initialization for a non-trivial rank-constrained problem (beyond PCA with full observations) with rank $r > 1$.

**Notation.** For matrices $\boldsymbol{X}, \boldsymbol{Y} \in \mathbb{R}^{n \times n}$, their inner product is $\langle \boldsymbol{X}, \boldsymbol{Y} \rangle = \operatorname{trace}\left(\boldsymbol{X}^\top \boldsymbol{Y}\right)$. We use $\|\boldsymbol{X}\|_F$, $\|\boldsymbol{X}\|_2$ and $\|\boldsymbol{X}\|_*$ for the Frobenius, spectral and nuclear norms of a matrix respectively. Given a matrix $\boldsymbol{X}$, we use $\sigma_i(\boldsymbol{X})$ to denote singular values of $\boldsymbol{X}$ in decreasing order. $\boldsymbol{X}_r = \arg\min_{\operatorname{rank}(\boldsymbol{Y}) \leq r} \|\boldsymbol{X} - \boldsymbol{Y}\|_F$ denotes the rank-$r$ approximation of $\boldsymbol{X}$, as obtained via its truncated singular value decomposition. We use plain capitals $R$ and $Q$ to denote orthonormal matrices.

## 2  Formulation and Assumptions

We write the linear measurement operator $\mathcal{A} : \mathbb{R}^{n \times n} \to \mathbb{R}^m$ as $\mathcal{A}(\boldsymbol{X})_i = \langle \boldsymbol{A}_i, \boldsymbol{X} \rangle$ where $\boldsymbol{A}_i \in \mathbb{R}^{n \times n}$, yielding $y_i = \langle \boldsymbol{A}_i, \boldsymbol{X}^* \rangle + w_i, i = 1, \cdots, m$. We assume $w_i \sim \mathcal{N}(0, \sigma_w^2)$ is i.i.d Gaussian noise. We are generally interested in the high dimensional regime where the number of measurements $m$ is usually much smaller than the dimension $n^2$.

Even if we know that $\operatorname{rank}(\boldsymbol{X}^*) \leq r$, having many measurements might not be sufficient for recovery if they are not "spread out" enough. E.g., if all measurements only involve the first $n/2$ rows and columns, we would never have any information on the bottom-right block. A sufficient condition for identifiability of a low-rank $\boldsymbol{X}^*$ from linear measurements by Recht et al. [20] is based on restricted isometry property defined below.

**Definition 2.1** (Restricted Isometry Property). *Measurement operator $\mathcal{A} : \mathbb{R}^{n \times n} \to \mathbb{R}^m$ (with rows $\boldsymbol{A}_i$, $i = 1, \cdots, m$) satisfies $(r, \delta_r)$ RIP if for any $n \times n$ matrix $\boldsymbol{X}$ with rank $\leq r$,*

$$(1 - \delta_r)\|\boldsymbol{X}\|_F^2 \leq \frac{1}{m}\sum_{i=1}^m \langle \boldsymbol{A}_i, \boldsymbol{X} \rangle^2 \leq (1 + \delta_r)\|\boldsymbol{X}\|_F^2. \tag{3}$$

In particular, $\boldsymbol{X}^*$ of rank $r$ is identifiable if $\delta_{2r} < 1$ [see 20, Theorem 3.2]. One situation in which RIP is obtained is for random measurement operators. For example, matrices with i.i.d. $\mathcal{N}(0, 1)$ entries satisfy $(r, \delta_r)$-RIP when $m = O(\frac{nr}{\delta^2})$ [see 6, Theorem 2.3]. This implies identifiability based on i.i.d. Gaussian measurement with $m = O(nr)$ measurements (coincidentally, the number of degrees of freedom in $\boldsymbol{X}^*$, optimal up to a constant factor).

## 3  Main Results

We are now ready to present our main result about local minima for the matrix sensing problem (2). We first present the results for noisy sensing of exact low rank matrices, and then generalize the results also to approximately low rank matrices.

Now we will present our result characterizing local minima of $f(\boldsymbol{U})$, for low-rank $\boldsymbol{X}^*$. Recall that measurements are $\boldsymbol{y} = \mathcal{A}(\boldsymbol{X}^*) + \boldsymbol{w}$, where entries of $\boldsymbol{w}$ are i.i.d. Gaussian - $w_i \sim \mathcal{N}(0, \sigma_w^2)$.

**Theorem 3.1.** *Consider the optimization problem (2) where $\boldsymbol{y} = \mathcal{A}(\boldsymbol{X}^*) + \boldsymbol{w}$, $\boldsymbol{w}$ is i.i.d. $\mathcal{N}(0, \sigma_w^2)$, $\mathcal{A}$ satisfies $(4r, \delta_{4r})$-RIP with $\delta_{4r} < \frac{1}{10}$, and $\operatorname{rank}(\boldsymbol{X}^*) \leq r$. Then, with probability $\geq 1 - \frac{10}{n^2}$ (over the noise), for any local minimum $\boldsymbol{U}$ of $f(\boldsymbol{U})$:*

$$\|\boldsymbol{U}\boldsymbol{U}^\top - \boldsymbol{X}^*\|_F \leq 20\sqrt{\frac{\log(n)}{m}}\sigma_w.$$

In particular, in the noiseless case ($\sigma_w = 0$) we have $\boldsymbol{U}\boldsymbol{U}^\top = \boldsymbol{X}^*$ and so $f(\boldsymbol{U}) = 0$ and every local minima is global. In the noiseless case, we can also relax the RIP requirement to $\delta_{4r} < 1/5$ (see Theorem 4.1 in Section 4). In the noisy case we cannot expect to ensure we always get to an exact global minima, since the noise might cause tiny fluctuations very close to the global minima possibly

creating multiple very close local minima. But we show that all local minima are indeed very close to some factorization $U^* U^{*\top} = X^*$ of the true signal, and hence to a global optimum, and this "radius" of local minima decreases as we have more observations.

The proof of the Theorem for the noiseless case is presented in Section 4. The proof for the general setting follows along the same lines and can be found in the Appendix.

So far we have discussed how all local minima are global, or at least very close to a global minimum. Using a recent result by Ge et al. [10] on the convergence of SGD for non-convex functions, we can further obtain a polynomial bound on the number of SGD iterations required to reach the global minima. The main condition that needs to be established in order to ensure this, is that all saddle points of (2) satisfy the "strict saddle point condition", i.e. have a direction with significant negative curvature:

**Theorem 3.2** (Strict saddle). *Consider the optimization problem* (2) *in the noiseless case, where* $y = \mathcal{A}(X^*)$, $\mathcal{A}$ *satisfies* $(4r, \delta_{4r})$*-RIP with* $\delta_{4r} < \frac{1}{10}$, *and* $\mathrm{rank}(X^*) \leq r$. *Let* $U$ *be a first order critical point of* $f(U)$ *with* $UU^\top \neq X^*$. *Then the smallest eigenvalue of the Hessian satisfies*

$$\lambda_{\min}\left[\frac{1}{m}\nabla^2(f(U))\right] \leq \frac{-2}{5}\sigma_r(X^*).$$

Now consider the stochastic gradient descent updates,

$$U^+ = \mathrm{Proj}_b\left(U - \eta\left(\sum_{i=1}^m (\langle A_i, UU^\top\rangle - y_i)A_i U + \psi\right)\right), \tag{4}$$

where $\psi$ is uniformly distributed on the unit sphere and $\mathrm{Proj}_b$ is a projection onto $\|U\|_F \leq b$. Using Theorem 3.2 and the result of Ge et al. [10] we can establish:

**Theorem 3.3** (Convergence from random initialization). *Consider the optimization problem* (2) *under the same noiseless conditions as in Theorem 3.2. Using* $b \geq \|U^*\|_F$, *for some global optimum* $U^*$ *of* $f(U)$, *for any* $\epsilon, c > 0$, *after* $T = poly\left(\frac{1}{\sigma_r(X^*)}, \sigma_1(X^*), b, \frac{1}{\epsilon}, \log(1/c)\right)$ *iterations of* (4) *with an appropriate stepsize* $\eta$, *starting from a random point uniformly distributed on* $\|U\|_F = b$, *with probability at least* $1 - c$, *we reach an iterate* $U_T$ *satisfying*

$$\|U_T - U^*\|_F \leq \epsilon.$$

The above result guarantees convergence of noisy gradient descent to a global optimum. Alternatively, second order methods such as cubic regularization (Nesterov and Polyak [19]) and trust region (Cartis et al. [7]) that have guarantees based on the strict saddle point property can also be used here.

**RIP Requirement:** Our results require $(4r, 1/10)$-RIP for the noisy case and $(4r, 1/5)$-RIP for the noiseless case. Requiring $(2r, \delta_{2r})$-RIP with $\delta_{2r} < 1$ is sufficient to ensure uniqueness of the global optimum of (1), and thus recovery in the noiseless setting [20], but all known efficient recovery methods require stricter conditions. The best guarantees we are aware of require $(5r, 1/10)$-RIP [20] or $(4r, 0.414)$-RIP [6] using a convex relaxation. Alternatively, $(6r, 1/10)$-RIP is required for global initialization followed by non-convex optimization [26]. In terms of requirements on $(2r, \delta_{2r})$-RIP for non-convex methods, the best we are aware of is requiring $\delta_{2r} < \Omega(1/r)$ [15, 29, 30]–this is a much stronger condition than ours, and it yields a suboptimal required number of spherical Gaussian measurements of $\Omega(nr^3)$. So, compared to prior work our requirement is very mild—it ensures efficient recovery, and requires the optimal number of spherical Gaussian measurements (up to a constant factor) of $O(nr)$.

**Extension to Approximate Low Rank** We can also obtain similar results that deteriorate gracefully if $X^*$ is not exactly low rank, but is close to being low-rank (see proof in the Appendix):

**Theorem 3.4.** *Consider the optimization problem* (2) *where* $y = \mathcal{A}(X^*)$ *and* $\mathcal{A}$ *satisfies* $(4r, \delta_{4r})$*-RIP with* $\delta_{4r} < \frac{1}{100}$, *Then, for any local minima* $U$ *of* $f(U)$:

$$\|UU^\top - X^*\|_F \leq 4(\|X^* - X_r^*\|_F + \delta_{2r}\|X^* - X_r^*\|_*),$$

*where* $X_r^*$ *is the best rank* $r$ *approximation of* $X^*$.

This theorem guarantees that any local optimum of $f(U)$ is close to $X^*$ upto an error depending on $\|X^* - X_r^*\|$. For the low-rank noiseless case we have $X^* = X_r^*$ and the right hand side vanishes. When $X^*$ is not exactly low rank, the best recovery error we can hope for is $\|X^* - X_r^*\|_F$, since $UU^\top$ is at most rank $k$. On the right hand side of Theorem 3.4, we have also a nuclear norm term, which might be higher, but it also gets scaled down by $\delta_{2r}$, and so by the number of measurements.

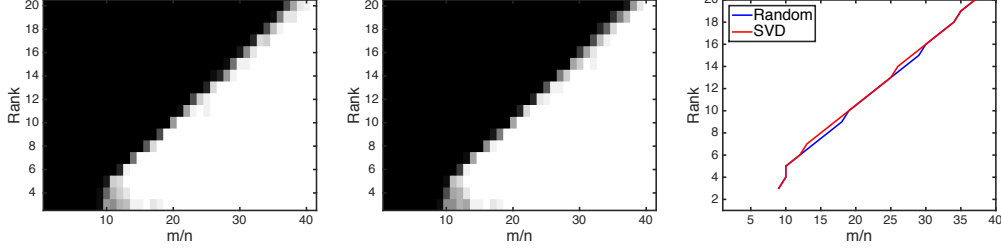

Figure 1: The plots in this figure compare the success probability of gradient descent between (left) random and (center) SVD initialization (suggested in [15]), for problem (2), with increasing number of samples $m$ and various values of rank $r$. Right most plot is the first $m$ for a given $r$, where the probability of success reaches the value 0.5. A run is considered success if $\|UU^\top - X^*\|_F / \|X^*\|_F \leq 1e-2$. White cells denote success and black cells denote failure of recovery. We set $n$ to be 100. Measurements $y_i$ are inner product of entrywise i.i.d Gaussian matrix and a rank-$r$ p.s.d matrix with random subspace. We notice no significant difference between the two initialization methods, suggesting absence of local minima as shown. Both methods have phase transition around $m = 2 \cdot n \cdot r$.

## 4 Proof for the Noiseless Case

In this section we present the proof characterizing the local minima of problem (2). For ease of exposition we first present the results for the noiseless case ($w = 0$). Proof for the general case can be found in the Appendix.

**Theorem 4.1.** *Consider the optimization problem* (2) *where* $y = \mathcal{A}(X^*)$, $\mathcal{A}$ *satisfies* $(4r, \delta_{4r})$-*RIP with* $\delta_{4r} < \frac{1}{5}$, *and* $\mathrm{rank}(X^*) \leq r$. *Then, for any local minimum* $U$ *of* $f(U)$:

$$UU^\top = X^*.$$

For the proof of this theorem we first discuss the implications of the first and second order optimality conditions and then show how to combine them to yield the result.

Invariance of $f(U)$ over $r \times r$ orthonormal matrices introduces additional challenges in comparing a given stationary point to a global optimum. We have to find the best orthonormal matrix $R$ to align a given stationary point $U$ to a global optimum $U^*$, where $U^*U^{*\top} = X^*$, to combine results from the first and second order conditions, without degrading the isometry constants.

Consider a local optimum $U$ that satisfies first and second order optimality conditions of problem (2). In particular $U$ satisfies $\nabla f(U) = 0$ and $z^\top \nabla^2 f(U) z \geq 0$ for any $z \in \mathbb{R}^{n \cdot r}$. Now we will see how these two conditions constrain the error $UU^\top - U^*U^{*\top}$.

First we present the following consequence of the RIP assumption [see 5, Lemma 2.1].

**Lemma 4.1.** *Given two* $n \times n$ *rank-$r$ matrices* $X$ *and* $Y$, *and a* $(4r, \delta)$-*RIP measurement operator* $\mathcal{A}$, *the following holds:*

$$\left| \frac{1}{m} \sum_{i=1}^m \langle A_i, X \rangle \langle A_i, Y \rangle - \langle X, Y \rangle \right| \leq \delta \|X\|_F \|Y\|_F. \tag{5}$$

### 4.1 First order optimality

First we will consider the first order condition, $\nabla f(U) = 0$. For any stationary point $U$ this implies

$$\sum_i \left\langle A_i, UU^\top - U^*U^{*\top} \right\rangle A_i U = 0. \tag{6}$$

Now using the isometry property of $\boldsymbol{A}_i$ gives us the following result.

**Lemma 4.2.** *[First order condition] For any first order stationary point $\boldsymbol{U}$ of $f(\boldsymbol{U})$, and $\mathcal{A}$ satisfying the $(4r, \delta)$-RIP (3), the following holds:*

$$\|(\boldsymbol{U}\boldsymbol{U}^\top - \boldsymbol{U}^*\boldsymbol{U}^{*\top})QQ^\top\|_F \leq \delta \left\|\boldsymbol{U}\boldsymbol{U}^\top - \boldsymbol{U}^*\boldsymbol{U}^{*\top}\right\|_F,$$

*where $Q$ is an orthonormal matrix that spans the column space of $\boldsymbol{U}$.*

This lemma states that any stationary point of $f(\boldsymbol{U})$ is close to a global optimum $\boldsymbol{U}^*$ in the subspace spanned by columns of $\boldsymbol{U}$. Notice that the error along the orthogonal subspace $Q_\perp$, $\|\boldsymbol{X}^* Q_\perp Q_\perp^\top\|_F$ can still be large making the distance between $\boldsymbol{X}$ and $\boldsymbol{X}^*$ arbitrarily far.

*Proof of Lemma 4.2.* Let $\boldsymbol{U} = QR$, for some orthonormal $Q$. Consider any matrix of the form $\boldsymbol{Z}QR^{\dagger\top}$ [2]. The first order optimality condition then implies,

$$\sum_{i=1}^m \left\langle \boldsymbol{A}_i, \boldsymbol{U}\boldsymbol{U}^\top - \boldsymbol{U}^*\boldsymbol{U}^{*\top}\right\rangle \left\langle \boldsymbol{A}_i, \boldsymbol{U}R^\dagger Q^\top \boldsymbol{Z}^\top\right\rangle = 0$$

The above equation together with Restricted Isometry Property (equation (5)) gives us the following inequality:

$$\left|\left\langle \boldsymbol{U}\boldsymbol{U}^\top - \boldsymbol{U}^*\boldsymbol{U}^{*\top}, QQ^\top \boldsymbol{Z}^\top\right\rangle\right| \leq \delta \left\|\boldsymbol{U}\boldsymbol{U}^\top - \boldsymbol{U}^*\boldsymbol{U}^{*\top}\right\|_F \left\|QQ^\top \boldsymbol{Z}^\top\right\|_F.$$

Note that for any matrix $\boldsymbol{A}$, $\langle \boldsymbol{A}, QQ^\top \boldsymbol{Z}\rangle = \langle QQ^\top \boldsymbol{A}, \boldsymbol{Z}\rangle$. Furthermore, for any matrix $\boldsymbol{A}$, $\sup_{\{\boldsymbol{Z}:\|\boldsymbol{Z}\|_F \leq 1\}} \langle \boldsymbol{A}, \boldsymbol{Z}\rangle = \|\boldsymbol{A}\|_F$. Hence the above inequality implies the lemma statement. $\square$

### 4.2 Second order optimality

We now consider the second order condition to show that the error along $Q_\perp Q_\perp^\top$ is indeed bounded well. Let $\nabla^2 f(\boldsymbol{U})$ be the hessian of the objective function. Note that this is an $n \cdot r \times n \cdot r$ matrix. Fortunately for our result we need to only evaluate the Hessian along $\mathrm{vec}(\boldsymbol{U} - \boldsymbol{U}^*R)$ for some orthonormal matrix $R$. Here $\mathrm{vec}(.)$ denotes writing a matrix in vector form.

**Lemma 4.3.** *[Hessian computation] Let $\boldsymbol{U}$ be a first order critical point of $f(\boldsymbol{U})$. Then for any $r \times r$ orthonormal matrix $R$ and $\Delta_j = \Delta e_j e_j^\top$ ( $\Delta = \boldsymbol{U} - \boldsymbol{U}^*R$),*

$$\sum_{j=1}^r \mathrm{vec}(\Delta_j)^\top \left[\nabla^2 f(\boldsymbol{U})\right] \mathrm{vec}(\Delta_j) = \sum_{i=1}^m \left(\sum_{j=1}^r 4 \left\langle \boldsymbol{A}_i, \boldsymbol{U}\Delta_j^\top\right\rangle^2 - 2 \left\langle \boldsymbol{A}_i, \boldsymbol{U}\boldsymbol{U}^\top - \boldsymbol{U}^*\boldsymbol{U}^{*\top}\right\rangle^2\right),$$

Hence from second order optimality of $\boldsymbol{U}$ we get,

**Corollary 4.1.** *[Second order optimality] Let $\boldsymbol{U}$ be a local minimum of $f(\boldsymbol{U})$ . For any $r \times r$ orthonormal matrix $R$,*

$$\sum_{j=1}^r \sum_{i=1}^m 4 \left\langle \boldsymbol{A}_i, \boldsymbol{U}\Delta_j^\top\right\rangle^2 \geq \frac{1}{2} \sum_{i=1}^m \left\langle \boldsymbol{A}_i, \boldsymbol{U}\boldsymbol{U}^\top - \boldsymbol{U}^*\boldsymbol{U}^{*\top}\right\rangle^2, \tag{7}$$

*Further for $\mathcal{A}$ satisfying $(2r, \delta)$ -RIP (equation (3)) we have,*

$$\sum_{j=1}^r \|\boldsymbol{U}e_j e_j^\top (\boldsymbol{U} - \boldsymbol{U}^*R)^\top\|_F^2 \geq \frac{1-\delta}{2(1+\delta)} \|\boldsymbol{U}\boldsymbol{U}^\top - \boldsymbol{U}^*\boldsymbol{U}^{*\top}\|_F^2. \tag{8}$$

The proof of this result follows simply by applying Lemma 4.3. The above Lemma gives a bound on the distance in the factor ($\boldsymbol{U}$) space $\|\boldsymbol{U}(\boldsymbol{U} - \boldsymbol{U}^*R)^\top\|_F^2$. To be able to compare the second order condition to the first order condition we need a relation between $\|\boldsymbol{U}(\boldsymbol{U} - \boldsymbol{U}^*R)^\top\|_F^2$ and $\|\boldsymbol{X} - \boldsymbol{X}^*\|_F^2$. Towards this we show the following result.

**Lemma 4.4.** *Let $U$ and $U^*$ be two $n \times r$ matrices, and $Q$ is an orthonormal matrix that spans the column space of $U$. Then there exists an $r \times r$ orthonormal matrix $R$ such that for any first order stationary point $U$ of $f(U)$, the following holds:*

$$\sum_{j=1}^{r} \|Ue_je_j^\top(U - U^*R)^\top\|_F^2 \leq \frac{1}{8}\|UU^\top - U^*U^{*\top}\|_F^2 + \frac{34}{8}\|(UU^\top - U^*U^{*\top})QQ^\top\|_F^2.$$

This Lemma bounds the distance in the factor space ($\|(U - U^*R)U^\top\|_F^2$) with $\|UU^\top - U^*U^{*\top}\|_F^2$ and $\|(UU^\top - U^*U^{*\top})QQ^\top\|_F^2$. Combining this with the result from second order optimality (Corollary 4.1) shows $\|UU^\top - U^*U^{*\top}\|_F^2$ is bounded by a constant factor of $\|(UU^\top - U^*U^{*\top})QQ^\top\|_F^2$. This implies $\|X^*Q_\perp Q_\perp\|_F$ is bounded, opposite to what the first order condition implied (Lemma 4.2). The proof of the above lemma is in Section B. Hence from the above optimality conditions we get the proof of Theorem 4.1.

*Proof of Theorem 4.1.* Assuming $UU^\top \neq U^*U^{*\top}$, from Lemmas 4.2, 4.4 and Corollary 4.1 we get,

$$\left(\frac{1-\delta}{2(1+\delta)} - \frac{1}{8}\right)\|UU^\top - U^*U^{*\top}\|_F^2 \leq \frac{34}{8}\delta^2 \left\|(UU^\top - U^*U^{*\top})\right\|_F^2.$$

If $\delta \leq \frac{1}{5}$ the above inequality holds only if $UU^\top = U^*U^{*\top}$. $\qquad\square$

## 5 Necessity of RIP

We showed that there are no spurious local minima only under a restricted isometry assumption. A natural question is whether this is necessary, or whether perhaps the problem (2) never has any spurious local minima, perhaps similarly to the non-convex PCA problem $\min_U \|A - UU^\top\|$.

A good indication that this is not the case is that (2) is NP-hard, even in the noiseless case when $y = \mathcal{A}(X^*)$ for $\mathrm{rank}(X^*) \leq k$ [20] (if we don't require RIP, we can have each $A_i$ be non-zero on a single entry in which case (2) becomes a matrix completion problem, for which hardness has been shown even under fairly favorable conditions [13])[3]. That is, we are unlikely to have a poly-time algorithm that succeeds for any linear measurement operator. Although this doesn't formally preclude the possibility that there are no spurious local minima, but it just takes a very long time to find a local minima, this scenario seems somewhat unlikely.

To resolve the question, we present an explicit example of a measurement operator $\mathcal{A}$ and $y = \mathcal{A}(X^*)$ (i.e. $f(X^*) = 0$), with $\mathrm{rank}(X^*) = r$, for which (1), and so also (2), have a non-global local minima.

*Example 1:* Let $f(X) = (X_{11} + X_{22} - 1)^2 + (X_{11} - 1)^2 + X_{12}^2 + X_{21}^2$ and consider (1) with $r = 1$ (i.e. a rank-1 constraint). For $X^* = \begin{bmatrix} 1 & 0 \\ 0 & 0 \end{bmatrix}$ we have $f(X^*) = 0$ and $\mathrm{rank}(X^*) = 1$. But $X = \begin{bmatrix} 0 & 0 \\ 0 & 1 \end{bmatrix}$ is a rank 1 local minimum with $f(X) = 1$.

We can be extended the construction to any rank $r$ by simply adding $\sum_{i=3}^{r+2}(X_{ii} - 1)^2$ to the objective, and padding both the global and local minimum with a diagonal beneath the leading $2 \times 2$ block.

In Example 1, we had a rank-$r$ problem, with a rank-$r$ exact solution, and a rank-$r$ local minima. Another question we can ask is what happens if we allow a larger rank than the rank of the optimal solution. That is, if we have $f(X^*) = 0$ with low $\mathrm{rank}(X^*)$, even $\mathrm{rank}(X^*) = 1$, but consider (1) or (2) with a high $r$. Could we still have non-global local minima? The answer is yes...

*Example 2:* Let $f(X) = (X_{11} + X_{22} + X_{33} - 1)^2 + (X_{11} - 1)^2 + (X_{22} - X_{33})^2 + \sum_{i,j:i\neq j} X_{ij}^2$ and consider the problem (1) with a rank $r = 2$ constraint. We can verify that $X^* = \begin{bmatrix} 1 & 0 & 0 \\ 0 & 0 & 0 \\ 0 & 0 & 0 \end{bmatrix}$ is a rank=1 global minimum with $f(X^*) = 0$, but $X = \begin{bmatrix} 0 & 0 & 0 \\ 0 & 1/2 & 0 \\ 0 & 0 & 1/2 \end{bmatrix}$ is a local minimum with

$f(\boldsymbol{X}) = 1$. Also for an arbitrary large rank constraint $r > 1$ (taking $r$ to be odd for simplicity), extend the objective to $f(\boldsymbol{X}) = (X_{11} - 1)^2 + \sum_{i=1}^{(r-1)/2} \left[ (X_{11} + X_{2i,2i} + X_{(2i+1),(2i+1)} - 1)^2 \right.$ $\left. + (X_{2i,2i} - X_{(2i+1),(2i+1)})^2 \right]$. We still have a rank-1 global minimum $\boldsymbol{X}^*$ with a single non-zero entry $X_{11}^* = 1$, while $\boldsymbol{X} = (I - \boldsymbol{X}^*)/2$ is a local minimum with $f(\boldsymbol{X}) = 1$.

## 6 Conclusion

We established that under conditions similar to those required for convex relaxation recovery guarantees, the non-convex formulation of matrix sensing (2) does not exhibit any spurious local minima (or, in the noisy and approximate settings, at least not outside some small radius around a global minima), and we can obtain theoretical guarantees on the success of optimizing it using SGD from *random initialization*. This matches the methods frequently used in practice, and can explain their success. This guarantee is very different in nature from other recent work on non-convex optimization for low-rank problems, which relied heavily on initialization to get close to the global optimum, and on local search just for the final local convergence to the global optimum. We believe this is the first result, together with the parallel work of Ge et al. [11], on the global convergence of local search for common rank-constrained problems that are worst-case hard.

Our result suggests that SVD initialization is not necessary for global convergence, and random initialization would succeed under similar conditions (in fact, our conditions are even weaker than in previous work that used SVD initialization). To investigate empirically whether SVD initialization is indeed helpful for ensuring global convergence, in Figure 1 we compare recovery probability of random rank-$k$ matrices for random and SVD initialization—there is no significant difference between the two.

Beyond the implications for matrix sensing, we are hoping these type of results could be a first step and serve as a model for understanding local search in deep networks. Matrix factorization, such as in (2), is a depth-two neural network with linear transfer—an extremely simple network, but already non-convex and arguably the most complicated network we have a good theoretical understanding of. Deep networks are also hard to optimize in the worst case, but local search seems to do very well in practice. Our ultimate goal is to use the study of matrix recovery as a guide in understating the conditions that enable efficient training of deep networks.

### Acknowledgements

Authors would like to thank Afonso Bandeira for discussions, Jason Lee and Tengyu Ma for sharing and discussing their work. This research was supported in part by an NSF RI/AF grant 1302662.

## Footnotes

[1]We study the case where $\boldsymbol{X}^*$ is PSD. We believe the techniques developed here can be used to extend results to the general case.

[2] $R^\dagger$ is the pseudo inverse of $R$

[3]Note that matrix completion is tractable under incoherence assumptions, similar to RIP

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
