[Supplementary Material · structure_lowrank_camera_appx.pdf]

# A Numerical Simulations

In this section we present simulation results for performance of gradient descent over $f(\boldsymbol{U})$. We consider measurements $y_i = \langle \boldsymbol{A}_i, \boldsymbol{X}^* \rangle$, where $\boldsymbol{A}_i$ are i.i.d Gaussian with each entry distributed as $\mathcal{N}(0, 1/m)$. $\boldsymbol{X}^*$ is a $100 \times 100$ rank $r$ random p.s.d matrix with $\|\boldsymbol{X}^*\|_F = 1$. $r$ is varied from 1 to 20 in the experiments.

We consider both standard gradient descent and noisy gradient descent (4) with step size $\frac{1}{\|\boldsymbol{U}\|_2}$. We add noise of magnitude $1e-4$ for the noisy gradient updates. Each method is run until convergence (max of 200 iterations). Let the output of gradient descent be $\widehat{\boldsymbol{U}}$. A run of this experiment is considered success if the final error $\|\widehat{\boldsymbol{U}}\widehat{\boldsymbol{U}}^\top - \boldsymbol{X}^*\|_F \le 1e-2$. Each experiment is repeated for 20 times and average probability of success is computed.

We repeat the above procedure starting from both random initialization and SVD initialization. For SVD initialization, the initial point is set to be the rank $r$ approximation of $\sum_{i=1}^m y_i \boldsymbol{A}_i$ as suggested by Jain et al. [15]. In figure 2 we have the plots for the cases discussed above. All of them have phase transition around number of samples $m = 2 \cdot n \cdot r$. This is in agreement with the results in Section 3. $f(\boldsymbol{U})$ has no local minima once $m \ge 2 \cdot n \cdot r$ and random initialization has same performance as SVD initialization.

In figure 3, the left two plots show error $\|\widehat{\boldsymbol{U}}\widehat{\boldsymbol{U}}^\top - \boldsymbol{X}^*\|_F / \|\boldsymbol{X}^*\|_F$ behaves with varying rank and number of samples for random and SVD initializations. The rightmost plot shows the phase transition for rank 10 case for all the methods. Again we notice no significant difference between these methods.

Figure 2: This figure plots the success probability for increasing number of samples $m$ and various values of rank $r$. The plots on the top are for gradient descent, left for random initialization and the right for SVD initialization. Similarly the bottom plots are for the noisy gradient descent. We notice no significant difference between all these settings. They all have phase transition around $m = 2 \cdot n \cdot r$.

Figure 3: This figure plots the error $\|\widehat{U}\widehat{U}^\top - X^*\|_F/\|X^*\|_F$ for increasing number of samples $m$. The left plot is for gradient descent with random initialization, center plot corresponds to SVD initialization. Again we notice no difference in error for these two settings. The rightmost figure shows phase transition of low rank recovery for all the different settings when $X^*$ is rank 10.

# B  Proof for the exact case

**Lemma** (4.3). *Let $U$ be a first order critical point of $f(U)$. Then for any $r \times r$ orthonormal matrix $R$ and $\Delta_j = \Delta e_j e_j^\top$ ( $\Delta = U - U^* R$),*

$$\sum_{j=1}^r vec(\Delta_j)^\top \left[\nabla^2 f(U)\right] vec(\Delta_j) = \sum_{i=1}^m (\sum_{j=1}^r 4 \left\langle A_i, U\Delta_j^\top \right\rangle^2 - 2 \left\langle A_i, UU^\top - U^*U^{*\top} \right\rangle^2),$$

*Proof of Lemma 4.3.* For any matrix $Z$, taking directional second derivative of the function $f(U)$ with respect to $Z$ we get:

$$vec(Z)^\top \left[\nabla^2 f(U)\right] vec(Z) = vec(Z)^\top \lim_{t \to 0} \left[\frac{\nabla f(U + t(Z)) - \nabla f(U)}{t}\right]$$

$$= 2\sum_{i=1}^m \left[2\left\langle A_i, UZ^\top \right\rangle^2 + \left\langle A_i, UU^\top - U^*U^{*\top} \right\rangle \left\langle A_i, ZZ^\top \right\rangle\right]$$

Setting $Z = \Delta_j = (U - U^* R)e_j e_j^\top$ and using the first order optimality condition on $U$, we get,

$$vec\left((U - U^* R)e_j e_j^\top\right)^\top \left[\nabla^2 f(U)\right] vec\left((U - U^* R)e_j e_j^\top\right)$$

$$= \sum_{i=1}^m 4\left\langle A_i, U\Delta_j^\top \right\rangle^2 + 2\left\langle A_i, UU^\top - U^*U^{*\top} \right\rangle \left\langle A_i, \Delta_j \Delta_j^\top \right\rangle$$

$$\overset{(i)}{=} \sum_{i=1}^m 4\left\langle A_i, Ue_j e_j^\top \Delta_j^\top \right\rangle^2 + 2\left\langle A_i, UU^\top - U^*U^{*\top} \right\rangle \left\langle A_i, U^* e_j e_j^\top (U^* e_j e_j^\top)^\top \right\rangle$$

$$\overset{(ii)}{=} \sum_{i=1}^m 4\left\langle A_i, Ue_j e_j^\top \Delta_j^\top \right\rangle^2 - 2\left\langle A_i, UU^\top - U^*U^{*\top} \right\rangle \left\langle A_i, Ue_j e_j^\top U^\top - U^* e_j e_j^\top U^{*\top} \right\rangle.$$

$(i)$ and $(ii)$ follow from the first order optimality condition (6),

$$\sum_{i=1}^m \left\langle A_i, UU^\top \right\rangle Ue_j e_j^\top = \sum_{i=1}^m \left\langle A_i, U^*U^{*\top} \right\rangle Ue_j e_j^\top,$$

for $j = 1 \cdots r$. Finally taking sum over $j$ from 1 to $r$ gives the result. $\qquad\square$

**Lemma** (4.4). *Let $U$ and $U^*$ be two $n \times r$ matrices, and $Q$ is an orthonormal matrix that spans the column space of $U$. Then there exists an $r \times r$ orthonormal matrix $R$ such that for any first order stationary point $U$ of $f(U)$, the following holds:*

$$\sum_{j=1}^r \|Ue_j e_j^\top (U - U^* R)^\top\|_F^2 \leq \frac{1}{8}\|UU^\top - U^*U^{*\top}\|_F^2 + \frac{34}{8}\|(UU^\top - U^*U^{*\top})QQ^\top\|_F^2.$$

*Proof of Lemma 4.4.* To prove this we will expand terms on the both sides in terms of $U$ and $\Delta = U - U^*R$ and then compare. First notice the following properties of $R$ that minimizes $\|U^*R - U\|_F$. Let $LSP^\top$ be the SVD of $U^{*\top}U$. Then, $R = LP^\top$. Hence, $R^\top U^{*\top}U = PSP^\top = U^\top U^*R$ is a PSD matrix. This implies, $U^\top \Delta = U^\top U - U^\top U^*R = U^\top U - R^\top U^{*\top}U = \Delta^\top U$.

Let columns of $U$ be orthogonal, else we can multiply $U$ by an orthonormal matrix and $UR$ will satisfy this. Since $UR$ is also local minimum, and $UU^\top = URR^\top U^\top$, results for $UR$ will also hold for $U$. Let $Q$ be the orthonormal matrix that spans the column space of $U$ and $Q_\perp Q_\perp^\top = I - QQ^\top$. Similarly let $Q_j$ span $Ue_je_j^\top$. Note that $Q_j$ are orthonormal since columns of $U$ are orthogonal. Hence,

$$\|(U - U^*R)e_je_j^\top U^\top\|_F^2 = \|Ue_je_j^\top U^\top - Q_jQ_j^\top U^*Re_je_j^\top U^\top - Q_{j\perp}Q_{j\perp}^\top U^*Re_je_j^\top U^\top\|_F^2$$
$$= \|Ue_je_j^\top U^\top - Q_jQ_j^\top U^*Re_je_j^\top U^\top\|_F^2 + \|Q_{j\perp}Q_{j\perp}^\top U^*Re_je_j^\top U^\top\|_F^2$$
$$\leq \frac{\|Ue_je_j^\top U^\top - Q_jQ_j^\top U^*Re_je_j^\top (Q_jQ_j^\top U^*R)^\top\|_F^2}{2(\sqrt{2}-1)} + \|Q_{j\perp}Q_{j\perp}^\top U^*Re_je_j^\top U^\top\|_F^2. \quad (9)$$

The last inequality follows from Lemma F.1 and the fact that $e_j^\top U^\top U^*Re_j \geq 0, \forall j$ as $U^\top U^*R$ is PSD. Now we will bound the second term in the above equation. The main idea here is to split this term into error between the subspaces of $X, X^*$ and then error between their singular values, since both of them are bounded by distance $\|X - X^*QQ^\top\|_F$. Let $Q^*$ be an orthonormal matrix that spans the column space of $X^*$. Also let $X = Q\Sigma_U^2 Q^\top$.

$$\|Q_{j\perp}Q_{j\perp}^\top U^*Re_je_j^\top U^\top\|_F^2 = \text{trace}(e_j^\top R^\top U^{*\top}Q_{j\perp}Q_{j\perp}^\top U^*Re_je_j^\top U^\top Ue_j)$$
$$= \text{trace}\left(e_j^\top R^\top U^{*\top}Q_{j\perp}Q_{j\perp}^\top U^*Re_j \left[e_j^\top U^\top Ue_j - e_j^\top R^\top U^{*\top}Q_jQ_j^\top Q_jQ_j^\top U^*Re_j\right.\right.$$
$$\left.\left.+e_j^\top R^\top U^{*\top}Q_jQ_j^\top U^*Re_j\right]\right)$$
$$\overset{(i)}{\leq} \frac{1}{8}\underbrace{(e_j^\top R^\top U^{*\top}Q_{j\perp}Q_{j\perp}^\top U^*Re_j)^2}_{\text{term1}} + 2\underbrace{(e_j^\top U^\top Ue_j - e_j^\top R^\top U^{*\top}Q_jQ_j^\top U^*Re_j)^2}_{\text{term2}}$$
$$+ \underbrace{(Q_{j\perp}Q_{j\perp}^\top U^*Re_je_j^\top (Q_jQ_j^\top U^*R)^\top)^2}_{\text{term3}}. \quad (10)$$

where $(i)$ follows from Cauchy-Schwarz inequality.

We will use the following inequality through the rest of the proof. So we state it first for any matrix $T$.

$$\sum_{j=1}^r (e_j^\top T^\top Te_j)^2 \leq \sum_{j=1}^r \sum_{k=1}^r (e_j^\top T^\top Te_k)^2$$
$$= \sum_{j=1}^r e_j^\top T^\top \left[\sum_{k=1}^r Te_ke_k^\top T^\top\right] Te_j = \sum_{j=1}^r e_j^\top T^\top TT^\top Te_j$$
$$= \|T^\top T\|_F^2 = \|TT^\top\|_F^2. \quad (11)$$

Now we will bound each of the terms in equation .
*Term 1:* Let, $T = Q_{j\perp}Q_{j\perp}^\top U^*R$. Then applying inequaltiy from equation (11) we get,

$$\sum_{j=1}^r (e_j^\top R^\top U^{*\top}Q_{j\perp}Q_{j\perp}^\top U^*Re_j)^2 = \sum_{j=1}^r (e_j^\top T^\top Te_j)^2$$
$$\leq \|T^\top T\|_F^2 = \|R^\top U^{*\top}Q_\perp Q_\perp^\top U^*R\|_F^2. \quad (12)$$

Further,

$$\|R^\top U^{*\top}Q_\perp Q_\perp^\top U^*R\|_F^2 = \text{trace}(U^{*\top}Q_\perp Q_\perp^\top U^*U^{*\top}Q_\perp Q_\perp^\top U^*)$$
$$= \text{trace}(Q_\perp Q_\perp^\top X^*Q_\perp Q_\perp^\top X^*)$$
$$\leq \|Q_\perp Q_\perp^\top X^*\|_F^2 \leq \|X - X^*\|_F^2. \quad (13)$$

*Term 2:*

$$(e_j^\top \boldsymbol{U}^\top \boldsymbol{U} e_j - e_j^\top R^\top \boldsymbol{U}^{*\top} Q_j Q_j^\top \boldsymbol{U}^* R e_j)^2$$

$$= (e_j^\top \boldsymbol{U}^\top \boldsymbol{U} e_j)^2 + (e_j^\top R^\top \boldsymbol{U}^{*\top} Q_j Q_j^\top \boldsymbol{U}^* R e_j)^2 - 2 e_j^\top \boldsymbol{U}^\top \boldsymbol{U} e_j e_j^\top R^\top \boldsymbol{U}^{*\top} Q_j Q_j^\top \boldsymbol{U}^* R e_j$$

$$= \|\boldsymbol{U} e_j e_j^\top \boldsymbol{U}^\top\|_F^2 + \|Q_j Q_j^\top \boldsymbol{U}^* R e_j e_j^\top R^\top \boldsymbol{U}^{*\top} Q_j Q_j^\top\|_F^2 - 2\operatorname{trace}(e_j^\top \boldsymbol{U}^\top \boldsymbol{U} e_j e_j^\top R^\top \boldsymbol{U}^{*\top} Q_j Q_j^\top \boldsymbol{U}^* R e_j)$$

$$\overset{(i)}{=} \|\boldsymbol{U} e_j e_j^\top \boldsymbol{U}^\top\|_F^2 + \|Q_j Q_j^\top \boldsymbol{U}^* R e_j e_j^\top R^\top \boldsymbol{U}^{*\top} Q_j Q_j^\top\|_F^2 - 2\operatorname{trace}(e_j^\top R^\top \boldsymbol{U}^{*\top} \boldsymbol{U} e_j e_j^\top \boldsymbol{U}^\top \boldsymbol{U}^* R e_j)$$

$$= \|\boldsymbol{U} e_j e_j^\top \boldsymbol{U}^\top - Q_j Q_j^\top \boldsymbol{U}^* R e_j e_j^\top R^\top \boldsymbol{U}^{*\top} Q_j Q_j^\top\|_F^2. \tag{14}$$

$(i)$ follows from $e_j^\top \boldsymbol{U}^\top \boldsymbol{U} e_j = \|\boldsymbol{U}_j\|_F^2$ and $\|\boldsymbol{U}_j\|_F^2 Q_j Q_j^\top = \boldsymbol{U} e_j e_j^\top \boldsymbol{U}^\top$. Now from orthogonality of $Q_j$ we have,

$$\sum_{j=1}^r \|\boldsymbol{U} e_j e_j^\top \boldsymbol{U}^\top - Q_j Q_j^\top \boldsymbol{U}^* R e_j e_j^\top R^\top \boldsymbol{U}^{*\top} Q_j Q_j^\top\|_F^2 \le \|\boldsymbol{U}\boldsymbol{U}^\top - QQ^\top \boldsymbol{U}^* \boldsymbol{U}^{*\top} QQ^\top\|_F^2. \tag{15}$$

*Term 3:* Finally we bound the last term in equation (10) similar to the first term, which gives,

$$\sum_{j=1}^r (Q_{j\perp} Q_{j\perp}^\top \boldsymbol{U}^* R e_j e_j^\top (Q_j Q_j^\top \boldsymbol{U}^* R)^\top)^2 \le \|\boldsymbol{U}\boldsymbol{U}^\top - \boldsymbol{U}^* \boldsymbol{U}^{*\top} QQ^\top\|_F^2.$$

Substituting the above equations (12), (13), (14) and (15) in (9) and (10) gives the result. □

# C   Proof for the Noisy Case

In this section we present the proof characterizing the local minima of problem (2). Recall $\boldsymbol{y} = \mathcal{A}(\boldsymbol{X}^*) + \boldsymbol{w}$, where $\boldsymbol{X}^*$ is a rank-$r$ matrix and $\boldsymbol{w}$ is i.i.d. $\mathcal{N}(0, \sigma_w^2)$.

We consider local optimum that satisfies first and second order optimality conditions of problem (2). In particular $\boldsymbol{U}$ satisfies $\nabla f(\boldsymbol{U}) = 0$ and $z^\top \nabla^2 f(\boldsymbol{U}) z \ge 0$ for any $z \in \mathbb{R}^{n \cdot r}$. Now we will see how these two conditions constrain the error $\boldsymbol{U}\boldsymbol{U}^\top - \boldsymbol{U}^* \boldsymbol{U}^{*\top}$.

## C.1   First order optimality

First we will consider the first order condition, $\nabla f(\boldsymbol{U}) = 0$. For any stationary point $\boldsymbol{U}$ this implies

$$\sum_i \left\langle \boldsymbol{A}_i, \boldsymbol{U}\boldsymbol{U}^\top - \boldsymbol{U}^* \boldsymbol{U}^{*\top} \right\rangle \boldsymbol{A}_i \boldsymbol{U} = \sum_{i=1}^m w_i \boldsymbol{A}_i \boldsymbol{U}. \tag{16}$$

Now using the isometry property of $\boldsymbol{A}_i$ gives us the following result.

**Lemma C.1.** *[First order condition] For any first order stationary point $\boldsymbol{U}$ of $f(\boldsymbol{U})$, and $\mathcal{A}$ satisfying the $(4r, \delta)$-RIP (3), the following holds:*

$$\|(\boldsymbol{U}\boldsymbol{U}^\top - \boldsymbol{U}^* \boldsymbol{U}^{*\top})QQ^\top\|_F \le \delta \left\|\boldsymbol{U}\boldsymbol{U}^\top - \boldsymbol{U}^* \boldsymbol{U}^{*\top}\right\|_F + 2\sqrt{\frac{(1+\delta)\log(n)}{m}}\sigma_w,$$

*w.p. $\ge 1 - \frac{1}{n^2}$, where $Q$ is an orthonormal matrix that spans the column space of $\boldsymbol{U}$.*

This lemma states that any stationary point of $f(\boldsymbol{U})$ is close to a global optimum $\boldsymbol{U}^*$ in the subspace spanned by columns of $\boldsymbol{U}$. Notice that the error along the orthogonal direction $\|\boldsymbol{X}^* Q_\perp Q_\perp^\top\|_F$ can still be large making the distance between $\boldsymbol{X}$ and $\boldsymbol{X}^*$ arbitrarily big.

*Proof of Lemma C.1.* Let $\boldsymbol{U} = QR$, for some orthonormal $Q$. Consider any matrix of the form $\boldsymbol{Z} Q R^{\dagger\top}$. The first order optimality condition then implies,

$$\sum_{i=1}^m \left\langle \boldsymbol{A}_i, \boldsymbol{U}\boldsymbol{U}^\top - \boldsymbol{U}^* \boldsymbol{U}^{*\top} \right\rangle \left\langle \boldsymbol{A}_i, \boldsymbol{U} R^\dagger Q^\top \boldsymbol{Z}^\top \right\rangle = \sum_{i=1}^m w_i \boldsymbol{A}_i \boldsymbol{U} R^\dagger Q^\top \boldsymbol{Z}^\top.$$

The above equation together with Restricted Isometry Property(equation (5)) gives us the following inequality:

$$\left|\left\langle \boldsymbol{UU}^\top - \boldsymbol{U}^*\boldsymbol{U}^{*\top}, QQ^\top \boldsymbol{Z}^\top \right\rangle\right| \le \delta \left\|\boldsymbol{UU}^\top - \boldsymbol{U}^*\boldsymbol{U}^{*\top}\right\|_F \|QQ^\top \boldsymbol{Z}^\top\|_F + 2\sqrt{\frac{(1+\delta)\log(n)}{m}} \sigma_w \|\boldsymbol{Z}^\top\|_F,$$

by Cauchy Schwarz inequality and Lemma F.2. Note that for any matrix $A$, $\left\langle \boldsymbol{A}, QQ^\top \boldsymbol{Z} \right\rangle = \left\langle \boldsymbol{A}QQ^\top, \boldsymbol{Z} \right\rangle$. Furthermore, for any matrix $A$, $\sup_{\{\boldsymbol{Z}:\|\boldsymbol{Z}\|_F \le 1\}} \langle \boldsymbol{A}, \boldsymbol{Z} \rangle = \|\boldsymbol{A}\|_F$. Hence the above inequality implies the lemma statement. $\square$

### C.2 Second order optimality

We will now consider the second order condition to show that the error along $Q_\perp Q_\perp^\top$ is indeed bounded well. Let $\nabla^2 f(\boldsymbol{U})$ be the hessian of the objective function. Note that this is an $n \cdot r \times n \cdot r$ matrix. Fortunately for our result we need to only evaluate the Hessian along the direction $\text{vec}(U - U^*R)$ for some orthonormal matrix $R$.

**Lemma C.2.** *[Hessian computation] Let $\boldsymbol{U}$ be a first order critical point of $f(\boldsymbol{U})$. Then for any $r \times r$ orthonormal matrix $R$ and $\Delta = \boldsymbol{U} - \boldsymbol{U}^*R$,*

$$\sum_{j=1}^{r} \text{vec}(\Delta_j)^\top \left[\nabla^2 f(\boldsymbol{U})\right] \text{vec}(\Delta_j)$$

$$= \sum_{i=1}^{m} \left( \sum_{j=1}^{r} 4 \left\langle \boldsymbol{A}_i, \boldsymbol{U}\Delta_j^\top \right\rangle^2 - 2 \left\langle \boldsymbol{A}_i, \boldsymbol{UU}^\top - \boldsymbol{U}^*\boldsymbol{U}^{*\top} \right\rangle^2 - 2w_i \left\langle \boldsymbol{A}_i, \boldsymbol{X} - \boldsymbol{X}^* \right\rangle \right),$$

*Proof of Lemma C.2.* For any matrix $\boldsymbol{Z}$, taking directional second derivative of the function $f(\boldsymbol{U})$ with respect to $\boldsymbol{Z}$ we get:

$$\text{vec}(\boldsymbol{Z})^\top \left[\nabla^2 f(\boldsymbol{U})\right] \text{vec}(\boldsymbol{Z}) = \text{vec}(\boldsymbol{Z})^\top \lim_{t\to 0} \left[\frac{\nabla f(\boldsymbol{U} + t(\boldsymbol{Z})) - \nabla f(\boldsymbol{U})}{t}\right]$$

$$= 2\sum_{i=1}^{m} \left[ 2\left\langle \boldsymbol{A}_i, \boldsymbol{UZ}^\top \right\rangle^2 + \left(\left\langle \boldsymbol{A}_i, \boldsymbol{UU}^\top - \boldsymbol{U}^*\boldsymbol{U}^{*\top} \right\rangle - w_i\right) \left\langle \boldsymbol{A}_i, \boldsymbol{ZZ}^\top \right\rangle \right]$$

Setting $\boldsymbol{Z} = \Delta_j = (\boldsymbol{U} - \boldsymbol{U}^*R)e_j e_j^\top$ and using the first order optimality condition on $\boldsymbol{U}$, we get,

$$\text{vec}\left((\boldsymbol{U} - \boldsymbol{U}^*R)e_j e_j^\top\right)^\top \left[\nabla^2 f(\boldsymbol{U})\right] \text{vec}\left((\boldsymbol{U} - \boldsymbol{U}^*R)e_j e_j^\top\right)$$

$$= \sum_{i=1}^{m} 4\left\langle \boldsymbol{A}_i, \boldsymbol{U}\Delta_j^\top \right\rangle^2 + 2\left(\left\langle \boldsymbol{A}_i, \boldsymbol{UU}^\top - \boldsymbol{U}^*\boldsymbol{U}^{*\top} \right\rangle - w_i\right)\left\langle \boldsymbol{A}_i, \Delta_j\Delta_j^\top \right\rangle$$

$$= \sum_{i=1}^{m} 4\left\langle \boldsymbol{A}_i, \boldsymbol{U}e_j e_j^\top \Delta_j^\top \right\rangle^2 + 2\left(\left\langle \boldsymbol{A}_i, \boldsymbol{UU}^\top - \boldsymbol{U}^*\boldsymbol{U}^{*\top} \right\rangle - w_i\right)\left\langle \boldsymbol{A}_i, \boldsymbol{U}^*e_j e_j^\top (\boldsymbol{U}^*e_j e_j^\top)^\top \right\rangle$$

$$= \sum_{i=1}^{m} 4\left\langle \boldsymbol{A}_i, \boldsymbol{U}e_j e_j^\top \Delta_j^\top \right\rangle^2 - 2\left\langle \boldsymbol{A}_i, \boldsymbol{UU}^\top - \boldsymbol{U}^*\boldsymbol{U}^{*\top} \right\rangle \left\langle \boldsymbol{A}_i, \boldsymbol{U}e_j e_j^\top \boldsymbol{U}^\top - \boldsymbol{U}^*e_j e_j^\top \boldsymbol{U}^{*\top} \right\rangle$$

$$- 2w_i \left\langle \boldsymbol{A}_i, \boldsymbol{U}e_j e_j^\top \boldsymbol{U}^\top - \boldsymbol{U}^*e_j e_j^\top \boldsymbol{U}^{*\top} \right\rangle. \tag{17}$$

where the last equality is again by the first order optimality condition (16). $\square$

Hence from second order optimality of $U$ we get,

**Corollary C.1.** *[Second order optimality] Let $U$ be a local minimum of $f(U)$ . For any $r \times r$ orthonormal matrix R, w.p. $\geq 1 - \frac{1}{n^2}$,*

$$\frac{1}{2}\sum_{i=1}^{m}\left\langle A_i, UU^\top - U^*U^{*\top}\right\rangle^2 \leq \sum_{i=1}^{m}\sum_{j=1}^{r}\left\langle A_i, U\Delta_j^\top\right\rangle^2 + \sqrt{\log(n)}\sigma_w\|\mathcal{A}(X - X^*)\|_2$$

$$\leq \sum_{i=1}^{m}\sum_{j=1}^{r}\left\langle A_i, U\Delta_j^\top\right\rangle^2 + 5\log(n)\sigma_w^2 + \frac{1}{20}\sum_{i=1}^{m}\left\langle A_i, X - X^*\right\rangle^2$$

*Further for $\mathcal{A}$ satisfying $(2r, \delta)$ -RIP (equation (3)) we have,*

$$\frac{1-\delta}{2(1+\delta)}\|UU^\top - U^*U^{*\top}\|_F^2 \leq \sum_{j=1}^{r}\|U\Delta_j^\top\|_F^2 + \frac{1}{20}\|X - X^*\|_F^2 + \frac{5\log(n)}{m(1+\delta)}\sigma_w^2. \quad (18)$$

Hence from the above optimality conditions we get the proof of Theorem 4.1.

*Proof of Theorem 3.1.* Assuming $UU^\top \neq U^*U^{*\top}$, from Lemma 4.4 and Corollary C.1 we get, with probability $\geq 1 - \frac{2}{n^2}$,

$$\left(\frac{1-\delta}{2(1+\delta)}\right)\|UU^\top - U^*U^{*\top}\|_F^2$$

$$\leq \frac{1}{8}\|X - X^*\|_F^2 + \frac{34}{8}\|X - X^*QQ^\top\|_F^2 + \frac{1}{20}\|X - X^*\|_F^2 + \frac{5\log(n)}{m(1+\delta)}\sigma_w^2$$

$$\overset{(i)}{\leq}\left(\frac{1}{8} + \frac{1}{20}\right)\|X - X^*\|_F^2 + \frac{34}{8}\left(2\delta^2\|X - X^*\|_F^2 + 8\frac{(1+\delta)\log(n)}{m}\sigma_w^2\right) + \frac{5\log(n)}{m(1+\delta)}\sigma_w^2.$$

$(i)$ follows from Lemma C.1. The above inequality implies,

$$\left(\frac{1-\delta}{2(1+\delta)} - \frac{1}{8} - \frac{1}{20} - \frac{34}{4}\delta^2\right)\|UU^\top - U^*U^{*\top}\|_F^2 \leq 34\frac{(1+\delta)\log(n)}{m}\sigma_w^2 + \frac{5\log(n)}{m(1+\delta)}\sigma_w^2.$$

If $\delta \leq \frac{1}{10}$, the above inequality reduces to $\|UU^\top - U^*U^{*\top}\|_F \leq c\sqrt{\frac{\log(n)}{m}}\sigma_w$, for some constant $c \leq 17$, w.p $\geq 1 - \frac{2}{n^2}$. $\qquad\square$

# D   Proof for the High Rank Case

In this section we will present the proof for the inexact case, where $\text{rank}(X^*) \geq r$. Recall that measurements are $y = \mathcal{A}(X^*)$.

Let SVD of $X^*$ be $Q^*\Sigma^*Q^*\top$. With slight abuse of notation we use $X^*_{jr+1:(j+1)r}$ to denote the $j$th rank $r$ block $Q^*_{jr+1:(j+1)r}\Sigma^*_{jr+1:(j+1)r}Q^*_{jr+1:(j+1)r}{}^\top$, where $Q^*_{jr+1:(j+1)r}$ denotes the restriction of $Q$ to columns $jr + 1$ to $(j + 1)r$.

## D.1   First order optimality

First we will consider the first order condition, $\nabla f(U) = 0$. For any stationary point $U$ this implies

$$\sum_i\left\langle A_i, UU^\top - U^*U^{*\top}\right\rangle A_iU = 0. \quad (19)$$

Now using the isometry property of $A_i$ gives us the following result.

**Lemma D.1.** *[First order condition] For any first order stationary point $U$ of $f(U)$, and $\{A_i\}$ satisfying the $(4r, \delta)$-RIP (3), the following holds:*

$$\|X - QQ^\top X^*_r\|_F \leq \delta\|X - X^*_r\|_F + \|(X^* - X^*_r)QQ^\top\|_F + \delta\|X^* - X^*_r\|_*.$$

*where $Q$ is an orthonormal matrix that spans the column space of $U$.*

This lemma states that any stationary point of $f(U)$ is close to a global optimum $U^*$ in the subspace spanned by columns of $U$. Notice that the error along the orthogonal direction $\|X^*Q_\perp Q_\perp^\top\|_F$ can still be large making the distance between $X$ and $X^*$ arbitrarily big.

*Proof of Lemma D.1.* Let $U = QR$, for some orthonormal $Q$. Consider any matrix of the form $ZQR^{\dagger \top}$. The first order optimality condition then implies,

$$\sum_{i=1}^m \langle A_i, X - X_r^* \rangle \langle A_i, UR^\dagger Q^\top Z^\top \rangle = \sum_{i=1}^m \langle A_i, X^* - X_r^* \rangle \langle A_i, UR^\dagger Q^\top Z^\top \rangle.$$

Note that $X - X_r^*$ is almost rank-$2r$. Hence, the above equation together with Restricted Isometry Property(equation (5)) gives us the following inequality:

$$\left| \langle X - X_r^*, QQ^\top Z^\top \rangle \right| - \delta \|X - X_r^*\|_F \|QQ^\top Z^\top\|_F$$
$$\leq \frac{1}{m} \sum_{i=1}^m \left\langle A_i, \sum_j X_{jr+1:(j+1)r}^* \right\rangle \langle A_i, QQ^\top Z^\top \rangle$$
$$\leq \sum_j \left\langle X_{jr+1:(j+1)r}^*, QQ^\top Z^\top \right\rangle + \delta \|X_{jr+1:(j+1)r}^*\|_F$$
$$\leq \|(X^* - X_r^*)QQ^\top\|_F + \delta \|X^* - X_r^*\|_*.$$

The last inequality follows from $\sum_j \|X_{jr+1:(j+1)r}^*\|_F \leq \|X^* - X_r^*\|_*$. The above inequalities are true for any $Z$.

Further note that for any matrix $A$, $\langle A, QQ^\top Z \rangle = \langle AQQ^\top, Z \rangle$. Furthermore, for any matrix $A$, $\sup_{\{Z: \|Z\|_F \leq 1\}} \langle A, Z \rangle = \|A\|_F$. Hence the above inequality implies the Lemma. $\square$

## D.2 Second order optimality

We will now consider the second order condition to show that the error along $Q_\perp Q_\perp^\top$ is indeed bounded well. Let $\nabla^2 f(U)$ be the hessian of the objective function. Note that this is an $n \cdot r \times n \cdot r$ matrix. Fortunately for our result we need to only evaluate the Hessian along the direction $\text{vec}(U - U^*R)$ for some orthonormal matrix $R$.

**Lemma D.2.** *[Hessian computation] Let $U$ be a first order critical point of $f(U)$. Then for any $n \times r$ matrix $Z$,*

$$\text{vec}(Z)^\top [\nabla^2 f(U)] \text{vec}(Z) = \sum_{i=1}^m 4 \langle A_i, UZ^\top \rangle^2 + 2 \langle A_i, UU^\top - U^*U^{*\top} \rangle \langle A_i, ZZ^\top \rangle,$$

*Further let $U$ be a local minimum of $f(U)$ and $\mathcal{A}$ satisfying $(2r, \delta)$ -RIP (equation (3)). Then,*

$$(1 - 3\delta)\|X - X_r^*\|_F^2 \leq 4(1 + \delta) \sum_{j=1}^r \|U\Delta_j^\top\|_F^2 + \|X^* - X_r^*\|_F^2 + \delta\|X^* - X_r^*\|_*^2.$$

*Proof of Lemma D.2.* For any matrix $Z$, taking directional second derivative of the function $f(U)$ with respect to $Z$ we get:

$$\text{vec}(Z)^\top [\nabla^2 f(U)] \text{vec}(Z) = \text{vec}(Z)^\top \lim_{t \to 0} \left[ \frac{\nabla f(U + t(Z)) - \nabla f(U)}{t} \right]$$
$$= 2 \sum_{i=1}^m \left[ 2 \langle A_i, UZ^\top \rangle^2 + \langle A_i, UU^\top - U^*U^{*\top} \rangle \langle A_i, ZZ^\top \rangle \right].$$

Setting $\boldsymbol{Z} = \Delta_j = (\boldsymbol{U} - \boldsymbol{U}^* R)e_j e_j^\top$ we get,

$$\sum_{j=1}^{r} \mathrm{vec}\left((\boldsymbol{U} - \boldsymbol{U}^* R)e_j e_j^\top\right)^\top \left[\nabla^2 f(\boldsymbol{U})\right] \mathrm{vec}\left((\boldsymbol{U} - \boldsymbol{U}^* R)e_j e_j^\top\right)$$

$$= \sum_{i=1}^{m}(\sum_{j=1}^{r} 4\left\langle \boldsymbol{A}_i, \boldsymbol{U}e_j e_j^\top (\boldsymbol{U} - \boldsymbol{U}_{\boldsymbol{r}}^* R)^\top\right\rangle^2$$

$$+ 2\sum_{j=1}^{r}\left\langle \boldsymbol{A}_i, \boldsymbol{U}\boldsymbol{U}^\top - \boldsymbol{U}^*\boldsymbol{U}^{*\top}\right\rangle\left\langle \boldsymbol{A}_i, (\boldsymbol{U} - \boldsymbol{U}_{\boldsymbol{r}}^* R)e_j e_j^\top (\boldsymbol{U} - \boldsymbol{U}_{\boldsymbol{r}}^* R)^\top\right\rangle)$$

$$\overset{(i)}{=} \sum_{i=1}^{m}(\sum_{j=1}^{r} 4\left\langle \boldsymbol{A}_i, \boldsymbol{U}\Delta_j^\top\right\rangle^2 + 2\left\langle \boldsymbol{A}_i, \boldsymbol{U}\boldsymbol{U}^\top - \boldsymbol{U}^*\boldsymbol{U}^{*\top}\right\rangle\left\langle \boldsymbol{A}_i, \boldsymbol{U}_{\boldsymbol{r}}^* R(\boldsymbol{U}_{\boldsymbol{r}}^* R)^\top - \boldsymbol{X}\right\rangle).$$

$(i)$ is by the first order optimality condition (19).

Hence from second order optimality of $U$ we get,

$$\sum_{i=1}^{m} 4\sum_{j=1}^{r}\left\langle \boldsymbol{A}_i, \boldsymbol{U}\Delta_j^\top\right\rangle^2 \geq \sum_{i=1}^{m} 2\left\langle \boldsymbol{A}_i, \boldsymbol{X} - \boldsymbol{X}^*\right\rangle\left\langle \boldsymbol{A}_i, \boldsymbol{X} - \boldsymbol{X}_{\boldsymbol{r}}^*\right\rangle. \qquad (20)$$

$$\frac{1}{m}\sum_{i=1}^{m}\left\langle \boldsymbol{A}_i, \boldsymbol{X} - \boldsymbol{X}^*\right\rangle\left\langle \boldsymbol{A}_i, \boldsymbol{X} - \boldsymbol{X}_{\boldsymbol{r}}^*\right\rangle = \frac{1}{m}\sum_{i=1}^{m}\left\langle \boldsymbol{A}_i, \boldsymbol{X} - \boldsymbol{X}_{\boldsymbol{r}}^*\right\rangle^2 + \left\langle \boldsymbol{A}_i, \boldsymbol{X}_{\boldsymbol{r}}^* - \boldsymbol{X}^*\right\rangle\left\langle \boldsymbol{A}_i, \boldsymbol{X} - \boldsymbol{X}_{\boldsymbol{r}}^*\right\rangle$$

$$\overset{(i)}{\geq} (1-\delta)\|\boldsymbol{X} - \boldsymbol{X}_{\boldsymbol{r}}^*\|_F^2 - \frac{1}{m}\sum_{i=1}^{m}\left(\sum_{j=1}\left\langle \boldsymbol{A}_i, \boldsymbol{X}_{jr+1:(j+1)r}^*\right\rangle\right)\left\langle \boldsymbol{A}_i, \boldsymbol{X} - \boldsymbol{X}_{\boldsymbol{r}}^*\right\rangle$$

$$\overset{(ii)}{\geq} (1-\delta)\|\boldsymbol{X} - \boldsymbol{X}_{\boldsymbol{r}}^*\|_F^2 - \sum_{j=1}\left\langle \boldsymbol{X} - \boldsymbol{X}_{\boldsymbol{r}}^*, \boldsymbol{X}_{jr+1:(j+1)r}^*\right\rangle - \delta\sum_{j=1}\|\boldsymbol{X} - \boldsymbol{X}_{\boldsymbol{r}}^*\|_F\|\boldsymbol{X}_{jr+1:(j+1)r}^*\|_F$$

$$= (1-\delta)\|\boldsymbol{X} - \boldsymbol{X}_{\boldsymbol{r}}^*\|_F^2 - \left\langle \boldsymbol{X} - \boldsymbol{X}_{\boldsymbol{r}}^*, \boldsymbol{X}^* - \boldsymbol{X}_{\boldsymbol{r}}^*\right\rangle - \delta\sum_{j=1}\|\boldsymbol{X} - \boldsymbol{X}_{\boldsymbol{r}}^*\|_F\|\boldsymbol{X}_{jr+1:(j+1)r}^*\|_F$$

$$\geq (1-\delta)\|\boldsymbol{X} - \boldsymbol{X}_{\boldsymbol{r}}^*\|_F^2 - \frac{1}{2}\|\boldsymbol{X} - \boldsymbol{X}_{\boldsymbol{r}}^*\|_F^2 - \frac{1}{2}\|\boldsymbol{X}^* - \boldsymbol{X}_{\boldsymbol{r}}^*\|_F^2 - \delta\sum_{j=1}\|\boldsymbol{X} - \boldsymbol{X}_{\boldsymbol{r}}^*\|_F\|\boldsymbol{X}_{jr+1:(j+1)r}^*\|_F$$

$$\overset{(iii)}{\geq} (1-\delta)\|\boldsymbol{X} - \boldsymbol{X}_{\boldsymbol{r}}^*\|_F^2 - \frac{1}{2}\|\boldsymbol{X} - \boldsymbol{X}_{\boldsymbol{r}}^*\|_F^2 - \frac{1}{2}\|\boldsymbol{X}^* - \boldsymbol{X}_{\boldsymbol{r}}^*\|_F^2 - \delta\frac{1}{2}\left(\|\boldsymbol{X} - \boldsymbol{X}_{\boldsymbol{r}}^*\|_F^2 + \|\boldsymbol{X}^* - \boldsymbol{X}_{\boldsymbol{r}}^*\|_*^2\right)$$

$$= \frac{1-3\delta}{2}\|\boldsymbol{X} - \boldsymbol{X}_{\boldsymbol{r}}^*\|_F^2 - \frac{1}{2}\|\boldsymbol{X}^* - \boldsymbol{X}_{\boldsymbol{r}}^*\|_F^2 - \frac{\delta}{2}\|\boldsymbol{X}^* - \boldsymbol{X}_{\boldsymbol{r}}^*\|_*^2. \qquad (21)$$

$(i)$ is from using RIP and splitting $\boldsymbol{X}^* - \boldsymbol{X}_{\boldsymbol{r}}^*$ into rank-$r$ components $\boldsymbol{X}^* - \boldsymbol{X}_{\boldsymbol{r}}^* = \sum_{j=1}^{n/r-1} \boldsymbol{X}_{jr+1:(j+1)r}^*$. $(ii)$ follows from using RIP (5). $(iii)$ follows from $\sum_j \|\boldsymbol{X}_{jr+1:(j+1)r}^*\|_F \leq \|\boldsymbol{X}^* - \boldsymbol{X}_{\boldsymbol{r}}^*\|_*$.

The Lemma now follows by combining equations (20), (21) and using RIP (3). $\qquad\square$

Hence from the above optimality conditions we get the proof of Theorem 3.4.

*Proof of Theorem 3.4.* Assuming $\boldsymbol{U}\boldsymbol{U}^\top \neq \boldsymbol{U}_{\boldsymbol{r}}^*\boldsymbol{U}_{\boldsymbol{r}}^{*\top}$, from Lemma 4.4 we know,

$$\sum_{j=1}^{r}\|\boldsymbol{U}\Delta_j^\top\|_F^2 \leq \frac{1}{8}\|\boldsymbol{U}\boldsymbol{U}^\top - \boldsymbol{U}_{\boldsymbol{r}}^*\boldsymbol{U}_{\boldsymbol{r}}^{*\top}\|_F^2 + \frac{34}{8}\|(\boldsymbol{U}\boldsymbol{U}^\top - \boldsymbol{U}_{\boldsymbol{r}}^*\boldsymbol{U}_{\boldsymbol{r}}^{*\top})QQ^\top\|_F^2, \qquad (22)$$

for some orthonormal $R$. Hence combining equations (22),with Lemma D.2 we get,

$$\frac{1-3\delta}{2}\|\boldsymbol{X} - \boldsymbol{X}_{\boldsymbol{r}}^*\|_F^2 \leq \frac{1}{2}\|\boldsymbol{X}^* - \boldsymbol{X}_{\boldsymbol{r}}^*\|_F^2 + \frac{\delta}{2}\|\boldsymbol{X}^* - \boldsymbol{X}_{\boldsymbol{r}}^*\|_*^2$$

$$+ 2(1+\delta)\left(\frac{1}{8}\|\boldsymbol{X} - \boldsymbol{X}_{\boldsymbol{r}}^*\|_F^2 + \frac{34}{8}\|(\boldsymbol{X} - \boldsymbol{X}_{\boldsymbol{r}}^*)QQ^\top\|_F^2\right).$$

This implies,

$$\frac{1-7\delta}{4}\|\boldsymbol{X}-\boldsymbol{X_r^*}\|_F^2 \le \frac{1}{2}\|\boldsymbol{X^*}-\boldsymbol{X_r^*}\|_F^2 + \frac{\delta}{2}\|\boldsymbol{X^*}-\boldsymbol{X_r^*}\|_*^2 + (1+\delta)\frac{17}{2}\|(\boldsymbol{X}-\boldsymbol{X_r^*})QQ^\top\|_F^2.$$
(23)

Finally from Lemma D.1 we know,

$$\|\boldsymbol{X}-\boldsymbol{X_r^*}QQ^\top\|_F^2 \le \left(\delta\|\boldsymbol{X}-\boldsymbol{X_r^*}\|_F + \|(\boldsymbol{X^*}-\boldsymbol{X_r^*})QQ^\top\|_F + \delta\|\boldsymbol{X^*}-\boldsymbol{X_r^*}\|_*\right)^2$$

$$\le \frac{11}{10}\|(\boldsymbol{X^*}-\boldsymbol{X_r^*})QQ^\top\|_F^2 + 22\delta^2\|\boldsymbol{X}-\boldsymbol{X_r^*}\|_F^2 + 22\delta^2\|\boldsymbol{X^*}-\boldsymbol{X_r^*}\|_*^2.$$
(24)

The last inequality follows from just using $2ab \le a^2 + b^2$.

Combining equations (23) and (24) gives,

$$\left(\frac{1-7\delta}{4} - \frac{17*22(1+\delta)\delta^2}{2}\right)\|\boldsymbol{X}-\boldsymbol{X_r^*}\|_F^2 \le \frac{1}{2}\|\boldsymbol{X^*}-\boldsymbol{X_r^*}\|_F^2 + \left(\frac{\delta}{2} + \frac{17*22\delta^2}{2}\right)\|\boldsymbol{X^*}-\boldsymbol{X_r^*}\|_*^2$$

$$+ (1+\delta)\frac{17*11}{20}\|(\boldsymbol{X^*}-\boldsymbol{X_r^*})QQ^\top\|_F^2$$

Substituting $\delta = \frac{1}{100}$ gives,

$$\|\boldsymbol{X}-\boldsymbol{X_r^*}\|_F^2 \le \frac{5}{2}\|\boldsymbol{X^*}-\boldsymbol{X_r^*}\|_F^2 + 12\delta\|\boldsymbol{X^*}-\boldsymbol{X_r^*}\|_*^2 + 10\|(\boldsymbol{X^*}-\boldsymbol{X_r^*})QQ^\top\|_F^2.$$

$$\le 13\|\boldsymbol{X^*}-\boldsymbol{X_r^*}\|_F^2 + 12\delta\|\boldsymbol{X^*}-\boldsymbol{X_r^*}\|_*^2.$$

$\square$

# E  Proofs for Section 3

In this section we present the proofs for the strict saddle theorem (Theorem 3.2) and the convergence guarantees (Theorem 3.3). The proofs use the Lemmas developed in Section 4 and the supporting Lemmas from Section F.

*Proof of Theorem 3.2.* From Lemma 4.3 we know that

$$\sum_{j=1}^r \text{vec}\,(\Delta_j)^\top \left[\frac{1}{m}\nabla^2 f(\boldsymbol{U})\right]\text{vec}\,(\Delta_j)$$

$$= \frac{1}{m}\sum_{i=1}^m (\sum_{j=1}^r 4\left\langle \boldsymbol{A}_i, \boldsymbol{U}\Delta_j^\top\right\rangle^2 - 2\left\langle \boldsymbol{A}_i, \boldsymbol{U}\boldsymbol{U}^\top - \boldsymbol{U^*}\boldsymbol{U^*}^\top\right\rangle^2$$

$$\le 4(1+\delta)\sum_{j=1}^r \|\boldsymbol{U}\Delta_j^\top\|_F^2 - 2(1-\delta)\|\boldsymbol{U}\boldsymbol{U}^\top - \boldsymbol{U^*}\boldsymbol{U^*}^\top\|_F^2,$$
(25)

where the last inequality follows from the RIP (3). Now applying Lemma 4.4 in equation (25) we get,

$$\sum_{j=1}^r \text{vec}\,(\Delta_j)^\top \left[\frac{1}{m}\nabla^2 f(\boldsymbol{U})\right]\text{vec}\,(\Delta_j)$$

$$\le (1+\delta)\left(\frac{1}{2}\|\boldsymbol{U}\boldsymbol{U}^\top - \boldsymbol{U^*}\boldsymbol{U^*}^\top\|_F^2 + 17\|(\boldsymbol{U}\boldsymbol{U}^\top - \boldsymbol{U^*}\boldsymbol{U^*}^\top)QQ^\top\|_F^2\right) - 2(1-\delta)\|\boldsymbol{U}\boldsymbol{U}^\top - \boldsymbol{U^*}\boldsymbol{U^*}^\top\|_F^2$$

$$= 17(1+\delta)\|(\boldsymbol{U}\boldsymbol{U}^\top - \boldsymbol{U^*}\boldsymbol{U^*}^\top)QQ^\top\|_F^2 - \frac{(3-5\delta)}{2}\|\boldsymbol{U}\boldsymbol{U}^\top - \boldsymbol{U^*}\boldsymbol{U^*}^\top\|_F^2$$

$$\overset{(i)}{\le} \left[17(1+\delta)\delta^2 - \frac{(3-5\delta)}{2}\right]\left\|\boldsymbol{U}\boldsymbol{U}^\top - \boldsymbol{U^*}\boldsymbol{U^*}^\top\right\|_F^2$$

$$\overset{(ii)}{\le} -1\cdot\left\|\boldsymbol{U}\boldsymbol{U}^\top - \boldsymbol{U^*}\boldsymbol{U^*}^\top\right\|_F^2.$$
(26)

$(i)$ follows from Lemma 4.2. $(ii)$ follows from $\delta \leq 1/10$. Now notice that from lemma F.1

$$\|\boldsymbol{X} - \boldsymbol{X}^*\|_F^2 \geq 2(\sqrt{2} - 1)\|(\boldsymbol{U} - \boldsymbol{U}^*R)(\boldsymbol{U}^*R)^\top\|_F^2$$
$$\geq 2(\sqrt{2} - 1)\sigma_r(\boldsymbol{X}^*)\|\boldsymbol{U} - \boldsymbol{U}^*R\|_F^2. \tag{27}$$

Finally notice that $\Delta_j = \Delta e_j e_j^\top$ have orthogonal columns. Hence,

$$\lambda_{\min}\left[\frac{1}{m}\nabla^2(f(\boldsymbol{U}, \boldsymbol{V}))\right] \leq \frac{1}{\|\boldsymbol{U} - \boldsymbol{U}^*R\|_F^2}\sum_{j=1}^r \mathsf{vec}\left(\Delta_j\right)^\top\left[\frac{1}{m}\nabla^2 f(\boldsymbol{U})\right]\mathsf{vec}\left(\Delta_j\right)$$

$$\overset{(i)}{\leq} \frac{-1}{\|\boldsymbol{U} - \boldsymbol{U}^*R\|_F^2}\left\|\boldsymbol{U}\boldsymbol{U}^\top - \boldsymbol{U}^*\boldsymbol{U}^{*\top}\right\|_F^2$$

$$\overset{(ii)}{\leq} \frac{-2(\sqrt{2} - 1)\sigma_r(\boldsymbol{X}^*)\|\boldsymbol{U} - \boldsymbol{U}^*R\|_F^2}{\|\boldsymbol{U} - \boldsymbol{U}^*R\|_F^2}$$

$$\leq \frac{-4}{5}\sigma_r(\boldsymbol{X}^*).$$

$(i)$ follows from equation (26). $(ii)$ follows from equation (27). $\qquad\square$

*Proof of Theorem 3.3.* To prove this theorem we use Theorem 6 of Ge et al. [10]. We need to show that $f(\boldsymbol{U})$ satisfies, 1) strict saddle property, 2) local strong convexity, 3) $f$ is bounded, smooth and has Lipschitz Hessian.

The boundedness assumption easily follows from assuming we are optimizing over a bounded domain $b$ such that, $\|\boldsymbol{U}^*\|_F \leq b$. Note that we can have any reasonable upper bound on the optimum and we can easily estimate this from $\sum_i y_i^2$ which is $\geq (1 - \delta)\|\boldsymbol{X}^*\|_F^2$ for the noiseless case.

Finally all the calculations below are for scaled version of $f(x)$ by $\frac{1}{m}$. Note that this does not change the number of iterations as both smoothness and strong convexity parameters are scaled by the same constant.

*Smoothness constant $\beta$:* Recall that smoothness of $f$ is bounded by maximum eigenvalue of Hessian over the domain. Hence, $\beta = \max_{\boldsymbol{Z}:\|\boldsymbol{Z}\|_F \leq 1} \boldsymbol{Z}^\top \nabla^2 f(\boldsymbol{U})\boldsymbol{Z}$. We have computed this projection of Hessian in Lemma C.2. Hence,

$$\beta = 2\max_{\boldsymbol{Z}:\|\boldsymbol{Z}\|_F^2 \leq 1}\sum_{i=1}^m\left[2\left\langle\boldsymbol{A}_i, \boldsymbol{U}\boldsymbol{Z}^\top\right\rangle^2 + \left\langle\boldsymbol{A}_i, \boldsymbol{U}\boldsymbol{U}^\top - \boldsymbol{U}^*\boldsymbol{U}^{*\top}\right\rangle\left\langle\boldsymbol{A}_i, \boldsymbol{Z}\boldsymbol{Z}^\top\right\rangle\right]$$

$$\overset{(i)}{\leq} \max_{\boldsymbol{Z}:\|\boldsymbol{Z}\|_F^2 \leq 1} 2\left(2(1 + \delta)\|\boldsymbol{U}\|_F^2\|\boldsymbol{Z}\|_F^2 + (1 + \delta)\|\boldsymbol{X} - \boldsymbol{X}^*\|_F\|\boldsymbol{Z}\boldsymbol{Z}^\top\|_F\right)$$

$$\leq 4(1 + \delta)b^2 + (1 + \delta)2b \leq 5b^2 + 3b.$$

$(i)$ follows from the RIP.

*$\rho$- Lipschitz Hessian:* Now we will compute the Lipschitz constant of Hessian of $f(\boldsymbol{U})$. We will first bound the spectral norm of difference of Hessian at two points $\boldsymbol{U}, \boldsymbol{V}$ in terms of $\|\boldsymbol{U} - \boldsymbol{V}\|_F$ along orthogonal direction $\boldsymbol{Z}_i$ and combine them to get bound on $\rho$.. Given two $n \times r$ matrices $\boldsymbol{U}, \boldsymbol{V}$,

$$\left\langle\nabla^2 f(\boldsymbol{U}) - \nabla^2 f(\boldsymbol{V}), \boldsymbol{Z}\boldsymbol{Z}^\top\right\rangle$$

$$\leq 2\max_{\boldsymbol{Z}:\|\boldsymbol{Z}\|_F^2 \leq 1}\sum_{i=1}^m\left[2\left\langle\boldsymbol{A}_i, \boldsymbol{U}\boldsymbol{Z}^\top\right\rangle^2 + \left\langle\boldsymbol{A}_i, \boldsymbol{U}\boldsymbol{U}^\top - \boldsymbol{U}^*\boldsymbol{U}^{*\top}\right\rangle\left\langle\boldsymbol{A}_i, \boldsymbol{Z}\boldsymbol{Z}^\top\right\rangle\right]$$

$$- \sum_{i=1}^m\left[2\left\langle\boldsymbol{A}_i, \boldsymbol{V}\boldsymbol{Z}^\top\right\rangle^2 + \left\langle\boldsymbol{A}_i, \boldsymbol{V}\boldsymbol{V}^\top - \boldsymbol{U}^*\boldsymbol{U}^{*\top}\right\rangle\left\langle\boldsymbol{A}_i, \boldsymbol{Z}\boldsymbol{Z}^\top\right\rangle\right]$$

$$\leq 4(1 + \delta)(\|\boldsymbol{U}\boldsymbol{Z}^\top\|_F^2 - \|\boldsymbol{V}\boldsymbol{Z}^\top\|_F^2) + 2(1 + \delta)\|\boldsymbol{U}\boldsymbol{U}^\top\boldsymbol{V}\boldsymbol{V}^\top\|_F\|\boldsymbol{Z}\boldsymbol{Z}^\top\|_F$$

$$\leq 4(1 + \delta)\|Z\|_F^2(\|\boldsymbol{U} - \boldsymbol{V}\|_F^2 + 2\|\boldsymbol{U}\|_F\|\boldsymbol{U} - \boldsymbol{V}\|_F) + 2(1 + \delta)\|\boldsymbol{U}\boldsymbol{U}^\top\boldsymbol{V}\boldsymbol{V}^\top\|_F$$

$$\leq \|Z\|_F^2\|\boldsymbol{U} - \boldsymbol{V}\|_F(8(1 + \delta)b + 4(1 + \delta)b)$$

$$= \|Z\|_F^2\|\boldsymbol{U} - \boldsymbol{V}\|_F(12(1 + \delta)b). \tag{28}$$

Hence, using the variational characterization of the Frobenius norm, the Hessian Lipschitz constant is bounded by $\max\{Z_i\}\sum_i \langle \nabla^2 f(U) - \nabla^2 f(V), Z_i Z_i^\top \rangle$, where $Z_i$ are orthogonal with $\sum_i \|Z_i\|_F^2 \leq 1$. Hence from equation (28) we get $\rho = O(b)$.

*Strict saddle property:* So far we have shown regularity properties of $f(U)$. Now we will discuss the strict saddle property. Theorem 3.2 shows that $\lambda_{\min}\left[\nabla^2(f(U))\right] \leq \frac{-2}{5}\sigma_r(X^*)$. To use results of [10] we need to show this property over an $\epsilon$ neighborhood of any saddle point $U$. For this first recall by smoothness, $\|\nabla f(U) - \nabla f(V)\|_F \leq \beta\|U - V\|_F$. Therefore $\nabla f(V) \leq \epsilon$, when $\|U - V\|_F \leq \frac{\epsilon}{\beta}$. Further we know the Hessian spectral norm is $\rho$ Lipschitz from equation (28). Hence, for any direction $Z$,

$$Z^\top \left(\nabla^2(f(V)) - \nabla^2(f(U))\right) Z^\top \leq \rho\|U - V\|_F \leq \rho\frac{\epsilon}{\beta}.$$

In particular choosing $Z$ to be the projection direction, $U - U^*$ implies from Theorem 3.2,

$$Z^\top \left(\nabla^2(f(V))\right) Z^\top \leq \frac{-2}{5}\sigma_r(X^*) + \rho\frac{\epsilon}{\beta}.$$

Hence for all $V$ in the bowl of radius $\epsilon$ around $U$, where $\epsilon \leq \frac{\beta}{5\rho}\sigma_r(X^*)$,

$$\lambda_{\min}\left[\nabla^2(f(V))\right] \leq \frac{-1}{5}\sigma_r(X^*). \tag{29}$$

*Local strong convexity:* Finally we need to show that the function is $\alpha$ strongly convex in a neighborhood $\theta$ around the optimum $U^*R$, for any orthonormal $R$. This easily follows from existing local convergence results for this problem. For example, Lemma 6.1 of Bhojanapalli et al. [2] states that, for $\|U - U^*R\|_F \leq \frac{\sigma_r(X^*)}{200\sigma_1(X^*)}\sigma_r(U^*R)$,

$$\langle \nabla f(U), U - U^*R \rangle \geq \frac{2}{3}\eta\|\nabla f(U)\|_F^2 + \frac{27}{200}\sigma_r(U^*R)^2\|U - U^*R\|_F^2. \tag{30}$$

for $\delta = \frac{1}{10}$ and some step size $\eta \propto \frac{1}{\|X^*\|_2}$. Hence $f(U)$ is locally strong convex with $\alpha = \frac{27}{200}\sigma_r(U^*R)^2$ in the neighborhood of radius $\theta = \frac{\sigma_r(X^*)}{200\sigma_1(X^*)}\sigma_r(U^*R)$ around the optimum.

Substituting these parameters in the Theorem 6 of Ge et al. [10] gives the result. □

# F Supporting Lemmas

In this section we present the supporting results used in the proofs above.

The following lemma relates the error $\|(U - Y)U^\top\|_F$ with $\|UU^\top - YY^\top\|_F$ under some conditions on $U$ and $Y$. This is a generalization of Lemma 5.4 in [26] and the proof follows similarly.

**Lemma F.1.** *Let $U$ and $Y$ be two $n \times r$ matrices. Further let $U^\top Y = Y^\top U$ be a PSD matrix. Then,*

$$\|(U - Y)U^\top\|_F^2 \leq \frac{1}{2(\sqrt{2} - 1)}\|UU^\top - YY^\top\|_F^2.$$

*Proof.* To prove this we will expand terms on the both sides in terms of $U$ and $\Delta = U - Y$ and then compare.

$$\|(\boldsymbol{U}\boldsymbol{U}^\top - \boldsymbol{Y}\boldsymbol{Y}^\top)\|_F^2 = \|(\boldsymbol{U}\Delta^\top + \Delta\boldsymbol{U}^\top - \Delta\Delta^\top)\|_F^2$$

$$= \operatorname{trace}\left(\Delta\boldsymbol{U}^\top\boldsymbol{U}\Delta^\top + \boldsymbol{U}\Delta^\top\Delta\boldsymbol{U}^\top + \Delta\Delta^\top\Delta\Delta^\top + 2\Delta\boldsymbol{U}^\top\Delta\boldsymbol{U}^\top - 2\Delta\Delta^\top\Delta\boldsymbol{U}^\top - 2\Delta\Delta^\top\boldsymbol{U}\Delta^\top\right)$$

$$\overset{(i)}{=} \operatorname{trace}\left(2\boldsymbol{U}^\top\boldsymbol{U}\Delta^\top\Delta + (\Delta^\top\Delta)^2 + 2(\boldsymbol{U}^\top\Delta)^2 - 4\Delta^\top\Delta\boldsymbol{U}^\top\Delta\right)$$

$$\overset{(ii)}{=} \operatorname{trace}\left(2\boldsymbol{U}^\top\boldsymbol{U}\Delta^\top\Delta + (\Delta^\top\Delta - \sqrt{2}\boldsymbol{U}^\top\Delta)^2 - 2(2 - \sqrt{2})\Delta^\top\Delta\boldsymbol{U}^\top\Delta\right)$$

$$\overset{(iii)}{\geq} 2\operatorname{trace}\left(\left[\boldsymbol{U}^\top\boldsymbol{U} - (2 - \sqrt{2})\boldsymbol{U}^\top\Delta\right]\Delta^\top\Delta\right)$$

$$= 2\operatorname{trace}\left(\left[(\sqrt{2} - 1)\boldsymbol{U}^\top\boldsymbol{U} + (2 - \sqrt{2})\boldsymbol{U}^\top\boldsymbol{Y}\right]\Delta^\top\Delta\right)$$

$$\overset{(iv)}{\geq} 2\operatorname{trace}\left((\sqrt{2} - 1)\boldsymbol{U}^\top\boldsymbol{U}\Delta^\top\Delta\right).$$

$(i)$ follows from the following properties of trace: $\operatorname{trace}(\boldsymbol{A}\boldsymbol{B}) = \operatorname{trace}(\boldsymbol{B}\boldsymbol{A})$ and $\operatorname{trace}(\boldsymbol{A}) = \operatorname{trace}(\boldsymbol{A}^\top)$. $(ii)$ follows from completing the squares. $(iii)$ follows from $\operatorname{trace}(\boldsymbol{A}^2) \geq 0$. $(iv)$ follows from the hypothesis of the lemma ($\boldsymbol{U}^\top\boldsymbol{Y}$ is PSD) and $\operatorname{trace}(\boldsymbol{A}\boldsymbol{B}) \geq 0$ for PSD matrices $\boldsymbol{A}$ and $\boldsymbol{B}$.

Finally notice that $\|(\boldsymbol{U} - \boldsymbol{Y})\boldsymbol{U}^\top\|_F^2 = \operatorname{trace}(\boldsymbol{U}^\top\boldsymbol{U}\Delta^\top\Delta)$. This completes the proof. $\qquad\square$

We recall the standard Gaussian random variable concentration here.

**Lemma F.2.** *Let $w_i \approx \mathcal{N}(0, \sigma_w)$, then*

$$\sum_{i=1}^m w_i x_i \leq 2\sqrt{\log(n)}\sigma_w\|\boldsymbol{x}\|,$$

*with probability $\geq 1 - \frac{1}{n^2}$.*

*Proof.* Recall $\mathbb{E}\left[e^{tw_i}\right] = e^{\sigma_w^2 t^2/2}$. Then by Markov's inequality, $P(\sum_{i=1}^m w_i x_i \geq c\|\boldsymbol{x}\|) \leq \frac{e^{\sigma_w^2\|\boldsymbol{x}\|^2 t^2/2}}{e^{tc\|\boldsymbol{x}\|}} \leq e^{-c^2/2\sigma_w^2}$, by setting $t = \frac{c}{\sigma_w^2\|\boldsymbol{x}\|}$. Choosing $c = 2\sqrt{\log(n)}\sigma_w$ completes the proof.

$\qquad\square$