[Reviews · NeurIPS 2016]

Reviewer 1

Summary

This paper proves that, for a certain class of matrix recovery problems, the non-convex factored formulation is nice: 1) for noiseless problems, all local minima are global 2) for noisy problems, all local minima are nearly as good as the global optimum. The paper also provides numerical evidence, and gives examples to show that there are local minima in some matrix recovery problems.

Qualitative Assessment

The authors have done a nice job showing, in yet another way, that low rank matrix recovery with a RIP is easy. To my knowledge, this is a new and interesting geometric fact about the problem. But the problem is extremely restrictive, and already has many good solutions. Furthermore, de Sa et al have already shown global convergence of SGD to the optimum from any initialization on this problem; while that did not constitute a proof that there were no local optima, it certainly makes it less surprising, and should be mentioned. In section 5, it would be worthwhile explicitly noting that the RIP case is not NP hard. And in the example, please do verify explicitly that [1 0; 0 0] is a local minimum; you haven't shown that there's no descent direction or path in rank 1 matrices from it to the global optimum. The bibliography should be cleaned up: eg, Grothendieck should be capitalized.

Confidence in this Review

2-Confident (read it all; understood it all reasonably well)


Reviewer 2

Summary

This paper studies low-rank recovery from randomized RIP measurements. Recently a few papers have shown that gradient descent on a natural nonconvex factorized objective converges to the global optimum (and hence recovers the low-rank matrix). These results typically holds in a neighborhood of the optimal solution and often require an additional spectral initialization step to ensure that the starting iterate lands in this neighborhood. This paper shows that all local minima are global so that as long as one can avoid saddles, it is possible to get global convergence guarantees. Based on Ge et. al. the authors show show random initialization+extra noise avoids saddles and hence recovers the globally optimal solution.

Qualitative Assessment

I think this a short, concise and well written paper. I read the paper in complete detail and believe it to be technically sound. In my view this is a must accept at NIPS. some minor comments that may help improve the paper: - lines 42-43 I agree that there is a disconnect but this comparison is not completely accurate because (1) the rate of convergence is different and (2) one could argue that in practice we don't really add noise in each iteration or may not want to run stochastic gradient so there still is a disconnect. The disconnect is no longer in the initialization but rather in the iterations. - It seems to me that using the local convergence of [26] the convergence rate of Theorem 3.3 can be improved. - lines 251-255: this is not really an accurate comment e.g. see work of Wright et. al. on phase retrieval.

Confidence in this Review

3-Expert (read the paper in detail, know the area, quite certain of my opinion)


Reviewer 3

Summary

This paper studied the problem of low-rank matrix recovery, and shows that when the observation operator satisfies a well-known property (Restricted Isometry Property), then this problem has only one local minimum, even though the problem itself is nonconvex. Because of this fact, simple algorithms such as gradient descent can obtain the global minimum. Compared with previous works on nonconvex optimization with theoretical guarantee, this work does not require a good initialization and therefore is much more applicable in practice.

Qualitative Assessment

Theorem 3.1 is an important result in nonconvex optimization and it is the first such work that I am aware of for matrix completion, and I think this paper should be published because of it. One suggestion is that, it would be great to run this nonconvex optimization algorithm in practice and check the convergence rate, to see whether it performs better/faster than the convex relaxation approach in practice (it is clearly faster in theoretically).

Confidence in this Review

2-Confident (read it all; understood it all reasonably well)


Reviewer 4

Summary

This paper extends a recent line of research regarding convergence of local search on the popular non-convex low-rank factorization trick. Recently, many authors have proven various local convergence results, and use spectral initialization methods to land within the radius of convergence. In practice, however, it has been observed that random initialization works just as well. This paper bridges the gap between this empirical observation and the theory, and argues that for RIP matrix sensing the geometry of the problem is such that all local min are global (or close to global in the noisy case) and all saddles are strict. This provides some nice theoretical arguments for why random initialization works as well as it does in practice.

Qualitative Assessment

This is a nice result. I am going to list a few nits I had about the paper as I read along. I think addressing some of these points would improve the presentation of the paper. 1. There are a few cases which are not covered by the results. For instance, strict-saddle in noisy case + local min are close to global in high rank, noisy case. A discussion about why these cases are not covered would be nice; I am assuming that it is not just straightforward modification of the current proof? 2. In practice, I believe that random init + gradient descent without noise is sufficient. Further, I think the rate of convergence remains linear, with the caveat that the linear rate improves in phases. Theorem 3.3 does not quite capture this behavior, by requiring both Gaussian noise added to every iteration and a norm projection operator. While I understand the theoretical reasons of this (so the authors can leverage the Ge et al. work), some discussion about deviation from practice would be a good addition to the paper. 3. The discussion about "RIP requirement" below Theorem 3.3 is a bit contrived. Some authors choose to optimize for convergence rates at the expense of the delta requirement. I would guess that in each of the works compared to, there is some wiggle room to lower the constant. Furthermore, I think the paper's requirement is actually 4r-RIP. For instance, when the authors lower bound the RHS of Line 206, Eq. (7) in Corollary 4.1, UU^T - U*(U*)^T is rank at most 2 (in general, without knowledge that UU^T = U*(U*)^T), which in order to apply Lemma 4.1 requires 4r-RIP. 4. I think the title of Section 5, "Necessity of RIP" is a bit misleading. The example given just demonstrates that the non-convex f can have local min not equal to global min. This is not surprising given, as indicated by the authors, how NP hard problems can be encoded in such an f. However, we know that RIP is only sufficient, and not necessary; the Ge et. al. paper cited by the authors shows that matrix completion has no bad local min, but such a measurement operator is not RIP.

Confidence in this Review

2-Confident (read it all; understood it all reasonably well)


Reviewer 5

Summary

In this paper the authors consider a low-rank matrix approximation problem where the low-rank constraint is enforced directly by working in the factorized space and optimizing over a matrix U to solve a problem of the form min_U |A(UU^T) –y |_F^2 where A() is a linear operator. Since working in the factorized space obviously makes the problem non-convex, the main contribution of the paper is to show under what conditions a local descent algorithm will converge to a global optimum, with the key requirements being that the linear operator, A(), needs to satisfy the restricted isometry property.

Qualitative Assessment

Prior work has shown that local descent is guaranteed to converge to a global minimum for the same problem under similar conditions. This prior work, however, relied on a careful initialization of the variables in order to guarantee their results, while the current work does not have this requirement and the variables can be initialized randomly. Additionally, the current work also achieves slightly better bounds on the necessary level of restricted isometry required by the linear operator than some of the prior work. While the contribution is thus in some sense an incremental improvement over this prior work, the authors clearly review the relevant literature and establish where their contribution falls relative to this work. In addition to the theoretical contribution, the authors also provide experimental evidence that performing local descent from random initializations successfully achieves the global optimum under the same conditions as using a specially chosen initialization (based on a SVD), which was necessary for results from the prior work. 1) Overall, the technical content of the paper appears to be well established, and I do not have many significant comments in that regard. I do, however, have a question regarding the proofs of some of the lemmas. First, the proof of Lemma 4.2 begins by requiring a factorization of U into U=QR where U has orthonormal columns and R is invertible; however, if U is rank-deficient it seems that such a factorization will not exist. Please discuss the correctness of Lemma 4.2 in the case where U is rank-deficient. 2) In the proof of Lemma 4.3, I did not work out the algebra in detail, but it would seem that in the equations following line 202 there should be terms that contain UR^T(U^*)^T and U^*RU^T that arise from the ZZ^T term. Where have these gone? In general, many of the proofs skip significant steps and would be much clearer if more of the intermediate steps are shown. 3) Given comment 2 I would suggest perhaps moving more the proofs into the supplement where they could be expanded and described in more detail. Then, the description of the experiments (which only appear in the supplement) can be moved into the main body of the paper.

Confidence in this Review

2-Confident (read it all; understood it all reasonably well)


Reviewer 6

Summary

This paper considers so called “matrix sensing” problem to exploit low-rank structure under clean and noisy scenarios. The authors argue that convex relaxation of the rank constraint, i.e., the trace-norm constraint, can be inefficient and unrealistic, which is why many literatures utilize the non-convex, factorized representation. The paper shows that all local minima of this problem are actually global or close to global optimum, under mild RIP requirements compared to prior works. For the practical optimization consideration, the authors give a proof that SGD can give a polynomial convergence guarantee to a global optimum with random initialization.

Qualitative Assessment

The guarantees that the paper provide can be quite beneficial for practical algorithms. There have been many similar attempts in the literature, and many of them provides similar bounds for the correct recovery. Overall, however, this paper provides nice new technical results for a fundamental problem that has large potential applications. My main concern is about the presentation of the paper. Some explanations are largely repeating, and the order of proofs and explanations is somewhat confusing to the readers (i.e., some lemmas are used before they are given.). These can be refined for better readability. (The authors' response have mostly alleviated the following doubts.) Lemma 4.4 is vital for many theorems in the paper, but I found some parts of the proof (somewhat) strange. For example, the inequality in (25) is said to be based on Cauchy-Schwarz inequality, (Is it also based on the arithmetic mean-geometric mean inequality?) but this particular choice seems a bit odd. How the inequalities in Lines 501-502 are derived (especially the trace part)? I'd like to see more explanations about these in the rebuttal and the final paper. Another doubt is about the examples in Section 5. Are they appropriate? In example 1, if we use the factorized representation UU^T, X = [0 0; 0 1] seems to be a saddle point, not a local minimum. Besides, even though there are two simulation results, it would be great to see some experimental results on real-world problems. There are many typos in the paper. Some are; - notational typos: lines 196, 340, 393 - line 189: ||X*Q_perp Q_perp^T||_F is not a direction (the same in line 215). - line 507: What is "the second inequality?"

Confidence in this Review

2-Confident (read it all; understood it all reasonably well)